# Second-order Optimization under Heavy-Tailed Noise: Hessian Clipping and Sample Complexity Limits

**Abdurakhmon Sadiev**[*]
KAUST[†]
Center of Excellence for Generative AI

**Peter Richtárik**
KAUST
Center of Excellence for Generative AI

**Ilyas Fatkhullin**
ETH Zurich, ETH AI Center,
Georgia Institute of Technology

## Abstract

Heavy-tailed noise is pervasive in modern machine learning applications, arising from data heterogeneity, outliers, and non-stationary stochastic environments. While second-order methods can significantly accelerate convergence in light-tailed or bounded-noise settings, such algorithms are often brittle and lack guarantees under heavy-tailed noise—precisely the regimes where robustness is most critical. In this work, we take a first step toward a theoretical understanding of second-order optimization under heavy-tailed noise. We consider a setting where stochastic gradients and Hessians have only bounded $p$-th moments, for some $p \in (1, 2]$, and establish tight lower bounds on the sample complexity of any second-order method. We then develop a variant of normalized stochastic gradient descent that leverages second-order information and provably matches these lower bounds. To address the instability caused by large deviations, we introduce a novel algorithm based on gradient and Hessian clipping, and prove high-probability upper bounds that nearly match the fundamental limits. Our results provide the first comprehensive sample complexity characterization for second-order optimization under heavy-tailed noise. This positions Hessian clipping as a robust and theoretically sound strategy for second-order algorithm design in heavy-tailed regimes.

## 1 Introduction

We consider the stochastic optimization problem

$$\min_{x \in \mathbb{R}^d} F(x), \qquad F(x) := \mathbb{E}_{\xi \sim \mathcal{D}}[f(x, \xi)], \tag{1}$$

with $F$ a smooth (but potentially nonconvex) objective, and $\xi$ a random variable drawn from an unknown distribution. While first-order methods are widely used in practice due to their simplicity and scalability, the application of second-order methods remains limited in stochastic settings. This is in stark contrast to their strong theoretical properties: second-order methods such as Newton's method [Bennett, 1916, Kantorovich, 1948], cubic regularization [Nesterov and Polyak, 2006, Nesterov, 2008], their variants for finite-sum minimization [Kovalev et al., 2019], quasi-Newton [Dennis and Moré, 1977], and adaptive second-order algorithms [Doikov et al., 2024] can offer provably faster convergence rates. Furthermore, complexity-theoretic results demonstrate that second-order methods can outperform first-order ones in noiseless (or finite sum) settings, often with moderate additional computational cost [Agarwal and Hazan, 2018, Arjevani et al., 2019, 2020] only.

---

[*]Corresponding author: abdurakhmon.sadiev@kaust.edu.sa

[†]King Abdullah University of Science and Technology, Thuwal, Saudi Arabia.

39th Conference on Neural Information Processing Systems (NeurIPS 2025).

**Second-order methods are highly sensitive to noise.** First-order stochastic optimization (FOSO) methods–such as stochastic gradient descent (SGD) and its adaptive variants–are well understood, both algorithmically and from a complexity–theoretic perspective. These methods enjoy optimal convergence guarantees under a broad class of noise models, including bounded variance and infinite variance regimes [Gower et al., 2019, Khaled and Richtárik, 2020, Yang et al., 2023, Wang et al., 2021, Fatkhullin et al., 2025]. In contrast, the development of second-order stochastic optimization (SOSO) methods has been significantly hampered by their extreme sensitivity to noise, particularly in Hessian estimation. Existing second-order methods suffer from both practical instability and overly restrictive theoretical assumptions. For instance, stochastic extensions of cubic regularization [Ghadimi et al., 2017, Tripuraneni et al., 2018] and its momentum variants [Chayti et al., 2024] require bounded noise assumptions, which are rarely satisfied in modern large-scale learning tasks. Other frameworks–such as trust-region methods [Arjevani et al., 2020], recursive momentum schemes [Tran and Cutkosky, 2021], and extrapolation-based approaches [Antonakopoulos et al., 2022, Agafonov et al., 2023]–relax this to bounded variance, but still fall short of handling more realistic heavy-tailed noise distributions. To the best of our knowledge, no existing SOSO method can operate reliably under noise conditions with unbounded variance, highlighting a fundamental gap in the current landscape of stochastic optimization.

**Heavy-tailed noise in gradients and Hessians.** In recent years, the first-order optimization literature has increasingly moved beyond the bounded variance assumption, adopting more general noise models that better reflect empirical observations in modern applications. A particularly influential framework is the *bounded central moment* condition ($p$-BCM), which assumes

$$\mathbb{E}\left[\|\nabla f(x, \xi) - \nabla F(x)\|^p\right] \leq \sigma^p, \qquad \text{for some } p \in (1, 2],$$

thereby allowing for heavy-tailed noise with unbounded variance. This model aligns well with empirical findings in deep learning and reinforcement learning (RL), where heavy-tailed gradient noise is often observed [Garg et al., 2021, Zhang et al., 2020, Ahn et al., 2024, Simsekli et al., 2019, Battash et al., 2024]. Algorithms designed for robustness under $p$-BCM noise, such as those employing gradient clipping or normalization, have demonstrated both strong empirical performance and rigorous convergence guarantees [Zhang et al., 2020, Hübler et al., 2025]. These developments suggest that heavy-tailed noise models are not only practically relevant, but also theoretically tractable, providing a principled foundation for algorithm design. Given that second-order methods are even more sensitive to noise than first-order ones, this naturally motivates the extension of such robustness principles to the second-order setting. In this work, we take this step and aim to

*develop a comprehensive theory of second-order stochastic optimization under heavy-tailed noise.*

Specifically, we assume access to unbiased stochastic gradients and Hessian-vector products, each satisfying a $p$-BCM-type condition:

$$\mathbb{E}\left[\|\nabla^2 f(x, \xi) - \nabla^2 F(x)\|_{\text{op}}^p\right] \leq \sigma_h^p, \qquad \text{for some } p \in (1, 2].$$

Our goal is to characterize the fundamental performance limits and develop practical algorithms for minimizing $F(x)$ in this setting, with a focus on obtaining guarantees for finding points with small gradient norm $\|\nabla F(x)\| \leq \varepsilon$, either in expectation or with high probability.[3]

**From gradient to Hessian clipping.** Gradient clipping has become a standard tool in modern machine learning, particularly due to its empirical success in stabilizing training under heavy-tailed noise and ill-conditioned objectives [Pascanu et al., 2013, Schulman et al., 2017]. Theoretically, it has been shown to offer robustness under relaxed smoothness and moment assumptions [Polyak and Tsypkin, 1979, Jakovetić et al., 2023], and enable high-probability guarantees with logarithmic (sub-Gaussian) dependence on the failure probability. The latter property of gradient clipping is in stark contrast to the classical linear algorithms, e.g., SGD, Momentum SGD, for which high probability lower bounds were recently established [Fatkhullin et al., 2025]. These properties are well-documented across both convex [Nazin et al., 2019, Gorbunov et al., 2020, Davis et al., 2021, Gorbunov et al., 2024b, Liu and Zhou, 2023, Gorbunov et al., 2024a, Puchkin et al., 2024, Armacki et al., 2023] and nonconvex [Sadiev et al., 2023, Nguyen et al., 2023, Cutkosky and Mehta, 2021,

---

[3] It would be interesting to extend our results to finding second-order stationary points [Nesterov and Polyak, 2006, Arjevani et al., 2020], but we leave this investigation for future work.

Hübler et al., 2025] regimes. Crucially, high-probability results are both theoretically and practically appealing as they guarantee the performance of individual runs rather than for the average-case behavior.

Despite these advances, the robustness enabled by gradient clipping remains largely confined to first-order methods. In the second-order setting, where both gradients and Hessians may be corrupted by heavy-tailed noise, comparable algorithmic tools are conspicuously lacking. The core difficulty is not just technical but conceptual: existing SOSO algorithms do not include mechanisms to suppress the influence of extreme noise in Hessian estimates. This leads us to a fundamental question for the design of robust second-order stochastic methods:

*Can we develop second-order algorithms with high-probability convergence guarantees under simultaneous heavy-tailed noise in both gradients and Hessians?*

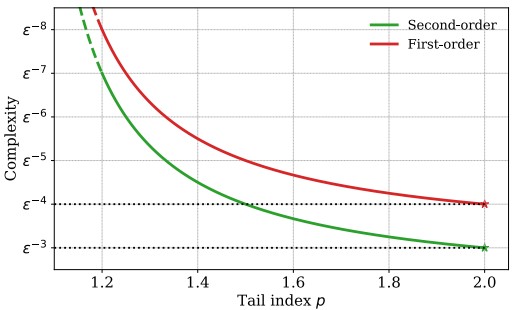

Figure 1: Sample complexity comparison for FOSO and SOSO depending on the tail index $p$. Each line corresponds to the leading term in the sample complexity for each class of algorithms. These leading terms match in upper and lower bounds, so this characterization is exact. We establish the characterization along the entire green line complementing prior work [Arjevani et al., 2023] for $p = 2$.

Figure 2: Performance of algorithms on a simple problem, $F(x) = 0.5 \|x\|^2$, $d = 10$ with synthetic noise generated from a two-sided Pareto distribution with tail index $p = 1.1$. We observe that algorithms without clipping, NSGDM and NSGDHess, suffer significantly from noise. This motivates our more in-depth study involving gradient and Hessian clipping for high probability convergence.

**Contributions:**

- **Tight lower bound.** We establish a minimax lower bound on the sample complexity of any SOSO algorithm under $p$-BCM noise model: $\Omega\left(\frac{\Delta \sigma_h}{\varepsilon^2}\left(\frac{\sigma}{\varepsilon}\right)^{\frac{1}{p-1}}\right)$, where $\Delta$ denotes the initial suboptimality, and $\sigma$, $\sigma_h$ denote the scale of noise in gradients and Hessians, respectively. This bound improves over the best known complexity for first-order methods [Zhang et al., 2020, Hübler et al., 2025] by a factor of $1/\varepsilon$ uniformly for all $p \in (1, 2]$, demonstrating that second-order methods can yield provable advantages even in the heavy-tailed regime; see Figure 1. Moreover, our result implies that increasing the order of the algorithm (e.g., using higher-order derivatives) cannot improve the rate under the same noise assumptions significantly.

- **Optimal algorithm.** We develop a second-order stochastic algorithm that achieves the lower bound (up to constants) in the regime $L \leq \sigma_h$, where $L$ denotes the Lipschitz constant of the gradient. The algorithm attains the sample complexity $\mathcal{O}\left(\frac{\Delta(L+\sigma_h)}{\varepsilon^2}\left(\frac{\sigma}{\varepsilon}\right)^{\frac{1}{p-1}}\right)$, and is, to the best of our knowledge, the first second-order method that provably works under $p$-BCM noise. Notably, our method does not require second-order smoothness of the objective. This result holds even in the classical setting $p = 2$, highlighting the broader implications of the approach.

- **High-probability convergence via Hessian clipping.** We propose a clipped variant of our algorithm that incorporates both gradient and *Hessian clipping*–the latter introduced here for the first time. We show that this variant achieves near-optimal sample complexity with high probability, incurring only a poly-logarithmic overhead in the failure probability. This

significantly extends the robustness of SOSO methods, enabling strong guarantees not only in expectation but also with high confidence.

Our results are primarily theoretical and validated through synthetic experiments with controllable heavy-tailed noise. While one component of our algorithm (in its unclipped form) has previously been applied in reinforcement learning settings [Fatkhullin et al., 2023], its broader practical deployment remains an open challenge. In particular, several questions must be addressed to make these methods viable in real-world applications: How can we reduce hyperparameter tuning overhead? Are there default configurations that are robust across problem classes? Is clipping truly necessary, or can it be systematically replaced by normalization techniques, as recent work suggests in the first-order context [Hübler et al., 2025]? We view our contributions as a foundational step towards a principled understanding of second-order methods under realistic, heavy-tailed noise conditions, and hope they spur further theoretical and algorithmic advances in this direction.

## 2 Notations and Assumptions

We use the common notation for natural numbers $\mathbb{N} = \{0, 1, \dots\}$, and let $[n] = \{1, 2, \dots, n\}$. Throughout this paper, $d \in \mathbb{N}_{\geq 1}$ denotes the dimension of the optimization problem (1). We use $\mathcal{O}(\cdot), \Omega(\cdot)$ for sample complexity notations, which preserve the dominating term in the target accuracy $\varepsilon > 0$ along with the multiplicative constants such as initial suboptimality $\Delta$, smoothness constants $L_r$, $r \geq 1$, the variance of higher-order stochastic oracles $\sigma_r$, $r \geq 1$. When we write $\bar{\mathcal{O}}(\cdot)$, we suppress the dependence on all parameters except for accuracy $\varepsilon$ in the dominating term. As in Carmon et al. [2020], for any $q$-th order tensor $T \in \mathbb{R}^{\otimes^q d}$, we define the support of $T$ as $\mathrm{supp}\{T\} \stackrel{\text{def}}{=} \{i \in [d] \mid T_i \neq 0\}$, where $T_i$ is the $(q-1)$-th order tensor, given by $[T_i]_{j_1, \dots, j_{q-1}} = T_{i, j_1, \dots, j_{q-1}}$.

We make the following assumptions throughout the paper.

**Assumption 1** (Lower Boundedness)**.** *The objective function $F$ is lower bounded by $F^* > -\infty$.*

**Assumption 2** ($L$-smoothness)**.** *The objective function $F$ is $L$-smooth, i.e., $F$ is differentiable and for all $x, y \in \mathbb{R}^d$, we have $\|\nabla F(x) - \nabla F(y)\| \leq L\|x - y\|$.*

**Assumption 3** ($p$-BCM)**.** *We have access to stochastic gradients $\nabla f(x, \xi)$ such that $\mathbb{E}[\nabla f(x, \xi)] = \nabla F(x)$. Moreover, there exist $p \in (1, 2]$ and $\sigma > 0$ such that*

$$\mathbb{E}[\|\nabla f(x, \xi) - \nabla F(x)\|^p] \leq \sigma^p \qquad \text{for all } x \in \mathbb{R}^d.$$

**Assumption 4** ($p$-BCM for Hessian)**.** *The function $F$ is twice differentiable and we additionally have access to stochastic Hessian-vector products $\nabla^2 f(x, \xi) \cdot v$ for any $v \in \mathbb{R}^d$ such that $\mathbb{E}[\nabla^2 f(x, \xi)] = \nabla^2 F(x)$. Moreover, there exist $p \in (1, 2]$ and $\sigma_h > 0$ such that*

$$\mathbb{E}\left[\|\nabla^2 f(x, \xi) - \nabla^2 F(x)\|_{op}^p\right] \leq \sigma_h^p \qquad \text{for all } x \in \mathbb{R}^d.$$

## 3 Lower Complexity Bounds for SOSO under $p$-BCM

To establish lower bounds, we build on a technique developed in a series of works [Arjevani et al., 2023, Carmon et al., 2020, Arjevani et al., 2020], which introduced a *worst-case* non-convex function exhibiting the zero-chain property: starting from the origin, each algorithmic iteration reveals information about a single coordinate only. Leveraging this framework, Zhang et al. [2020] derived the first lower bounds for FOSO under heavy-tailed noise, assuming the $p$-th moment of the stochastic gradient is bounded, i.e., $\mathbb{E}[\|\nabla f(x, \xi)\|^p] \leq G^p$, which differs from Assumption 3.

In our work, we assume access to a $q$-th order stochastic oracle, meaning that each query yields stochastic approximations of derivatives of orders 1 to $q$ at a given point $x$. This oracle model was formally introduced by Arjevani et al. [2020]. Building on their proof technique, we derive new lower bounds under the assumption that each stochastic derivative has a bounded $p$-th central moment.

**Function class.** We consider the class of $q$-times differentiable functions ($q \geq 1$) satisfying two key properties: (i) functional boundedness, i.e., $F(0) - F^* \leq \Delta$; and (ii) Lipschitz continuity of each derivative up to order $q$, i.e., for all $x, y \in \mathbb{R}^d$ and $r \in [q]$,

$$\|\nabla^r F(x) - \nabla^r F(y)\|_{\mathrm{op}} \leq L_r \|x - y\|.$$

We denote this function class by $\mathcal{F}(\Delta, L_{1:q})$, where $L_{1:q} = \{L_1, L_2, \ldots, L_q\}$. Note that the constant $L$ from Assumption 2 corresponds to $L_1$ in this notation.

**Oracle class.** We now define the oracle model. For a fixed function $F \in \mathcal{F}(\Delta, L_{1:q})$, a stochastic $q$-th order oracle is defined as follows: for any $x \in \mathbb{R}^d$ and $\xi \sim \mathcal{D}$,

$$\mathrm{O}_f^q(x, \xi) \stackrel{\text{def}}{=} \left( f(x, \xi), \nabla f(x, \xi), \nabla^2 f(x, \xi), \ldots, \nabla^q f(x, \xi) \right),$$

where each component satisfies the unbiasedness conditions: $\mathbb{E}_{\xi \sim \mathcal{D}}[f(x, \xi)] = F(x)$, and $\mathbb{E}_{\xi \sim \mathcal{D}}[\nabla^r f(x, \xi)] = \nabla^r F(x), \quad \forall\, r \in [q]$. We define the oracle class $\mathcal{O}_q(F, \sigma_{1:q})$ as the set of all such stochastic oracles for which the $p$-th moment of the estimation error (with $p \in [1, 2)$) satisfies

$$\mathbb{E}_{\xi \sim \mathcal{D}} \left[ \|\nabla^r f(x, \xi) - \nabla^r F(x)\|_{\mathrm{op}}^p \right] \leq \sigma_r^p, \quad \forall\, r \in [q], \ \forall\, x \in \mathbb{R}^d.$$

For $r = 1$, this recovers $\sigma_1 = \sigma$ from Assumption 3, and for $r = 2$, we similarly have $\sigma_2 = \sigma_h$ from Assumption 4.

**Algorithm class.** To prove lower bounds, we will work with a class of *zero-respecting* algorithms. The formal definition is provided below. The reason why this class of algorithms is interesting for us is based on its property: at each iteration of such an algorithm, at most one new coordinate can be revealed.

**Definition 1.** *We call a stochastic $q$-th order algorithm $A$ zero-respecting if for any function $F$ and any $p$-th order oracle $O_F^q$, the iterates $\{x^t\}_{t \in \mathbb{N}}$ generated by $A$ via querying $O_F^q$ satisfy the following property:* $\mathrm{supp}\,\{x^t\} \subset \bigcup_{i < t} \mathrm{supp}\,\{O_F^q(x^i, \xi^i)\}, \quad \forall\, t \in \mathbb{N}$ *with probability one with respect to the randomness of the algorithm and the realizations of $\{\xi^t\}_{t \in \mathbb{N}}$.*

**Theorem 1.** *Let $q \in \mathbb{N}_{\geq 1}$, and let $\Delta > 0$, $L_{1:q} \stackrel{\text{def}}{=} (L_1, \ldots, L_q)$, $\sigma_{1:q} \stackrel{\text{def}}{=} (\sigma_1, \ldots, \sigma_q)$ and $\varepsilon \leq \mathcal{O}(\sigma_1)$. Then, there exists $F \in \mathcal{F}(\Delta, L_{1:q})$ and a corresponding noisy oracle $O_F^q \in \mathcal{O}(F, \sigma_{1:q})$ such that for any $q$-th order zero-respecting algorithm, the number of oracle queries required to find an $\varepsilon$-stationary point (with constant probability) is lower bounded by*

$$\Omega(1) \cdot \frac{\Delta}{\varepsilon} \left( \frac{\sigma_1}{\varepsilon} \right)^{\frac{p}{p-1}} \min \left\{ \min_{r \in \{2, \ldots, q\}} \left( \frac{\sigma_r}{\sigma_1} \right)^{\frac{1}{r-1}}, \min_{r' \in [q]} \left( \frac{L_{r'}}{\varepsilon} \right)^{\frac{1}{r'}} \right\}. \tag{2}$$

*Moreover, this lower bounds is realized by a construction of dimension* $\Theta \left( \frac{\Delta}{\varepsilon} \min \left\{ \min_{r \in \{2, \ldots, q\}} \left( \frac{\sigma_r}{\sigma_1} \right)^{\frac{1}{r-1}}, \min_{r' \in [q]} \left( \frac{L_{r'}}{\varepsilon} \right)^{\frac{1}{r'}} \right\} \right).$

The proof of Theorem 1 is deferred to Appendix D. In Theorem 1 we provide lower bounds for any $q \in \mathbb{N}_{\geq 1}$, which corresponds to the order of an oracle used in an optimization algorithm. When the oracle order satisfies $q \geq 2$, the algorithm gains access to stochastic curvature information, enabling more accurate prediction of the function or gradient behavior between consecutive iterates. This richer information can potentially lead to tighter worst-case complexity limits compared to first-order methods. Since the main focus of our work is on the second-order setting, we now discuss the complexity result from Theorem 1 for $q = 2$. For second-order methods, $q = 2$, under Assumptions 2, 3, 4 and, additionally, (possibly) assuming the Hessian is Lipschitz continuous with constant $L_h$, we have

$$\Omega(1) \cdot \min \left\{ \frac{\Delta \sigma_h}{\varepsilon^2} \left( \frac{\sigma}{\varepsilon} \right)^{\frac{1}{p-1}}, \frac{\Delta L \sigma}{\varepsilon^3} \left( \frac{\sigma}{\varepsilon} \right)^{\frac{1}{p-1}}, \frac{\Delta \sqrt{L_h} \sigma}{\varepsilon^{5/2}} \left( \frac{\sigma}{\varepsilon} \right)^{\frac{1}{p-1}} \right\}, \tag{3}$$

where the parameters satisfy $\sigma = \sigma_1$, $\sigma_h = \sigma_2$, $L = L_1$, $L_2 = L_h$. We will now discuss each term separately, in different regimes.

**First-order optimization with Lipschitz gradient.** When we do not have access to the second-order information and the objective function has Lipschitz continuous gradient only, but not any higher derivative, our lower bounds (3) reduces to the second term:

$$\Omega \left( \frac{\Delta L \sigma}{\varepsilon^3} \left( \frac{\sigma}{\varepsilon} \right)^{\frac{1}{p-1}} \right).$$

This bound exactly matches (up to a numerical constant factor) the previously known lower bounds for first-order stochastic optimization under $p$-BCM [Zhang et al., 2020], and the corresponding upper bound achieved by normalized SGD [Hübler et al., 2025]. Several other algorithms, like Clip-SGD and NSGD with momentum and clipping also nearly match this complexity lower bound, up to additional polylogarithmic terms in $1/\varepsilon$ and polynomial terms in $\delta$, $L$ and $\sigma$ [Zhang et al., 2020, Cutkosky and Mehta, 2021, Nguyen et al., 2023].

**First-order optimization under higher-order smoothness.** The second case worth discussing is when we (still) do not have access to stochastic Hessian, but we know that the objective function has Lipschitz continuous second-order derivatives. Then our lower bound (3) becomes

$$\Omega\left(\min\left\{\frac{\Delta L\sigma}{\varepsilon^3}\left(\frac{\sigma}{\varepsilon}\right)^{\frac{1}{p-1}}, \frac{\Delta\sqrt{L_h}\sigma}{\varepsilon^{5/2}}\left(\frac{\sigma}{\varepsilon}\right)^{\frac{1}{p-1}}\right\}\right).$$

The last term in the above bound nearly matches with the complexity of the method called NIGT with clipping, by Cutkosky and Mehta [2021], in terms of its dependence on $1/\varepsilon$, up to logarithmic factors. Unfortunately, there is still a small discrepancy with the upper bound in the dependence on other parameters and with the fact that Cutkosky and Mehta [2021] use a slightly stronger (non-central) assumption $\mathbb{E}\left[\|\nabla f(x, \xi)\|^p\right] \leq G^p$ instead of our $p$-BCM. We believe our bound is tight and can be achieved with a more careful analysis of NIGT type algorithm under the $p$-BCM assumption.

**Second-order optimization with only Lipschitz gradient.** Finally, we wish to discuss an important setting, which is rarely discussed in the literature on second-order optimization. Higher-order smoothness can be challenging to verify in practice. Moreover, even when it is possible, the estimates of $L_h$ can be too large to be useful. Thus we pay attention to the case when we have access to stochastic Hessian, but the second derivative can be non-continuous, i.e., $L_h \to +\infty$. Our lower bound (3) provides new insights on this interesting setting, simplifying to

$$\Omega\left(\min\left\{\frac{\Delta\sigma_h}{\varepsilon^2}\left(\frac{\sigma}{\varepsilon}\right)^{\frac{1}{p-1}}, \frac{\Delta L\sigma}{\varepsilon^3}\left(\frac{\sigma}{\varepsilon}\right)^{\frac{1}{p-1}}\right\}\right).$$

As we can see, the first term corresponding to the second-order information has better dependence on an accuracy $\varepsilon$, and can be potentially much smaller than the second term. Moreover, this complexity is not impacted with the constants corresponding to higher-order smoothness, which potentially gives us the possibility of designing second-order methods with complexity that does not depend on $L_h$ and higher order smoothness constants. However, this is merely a lower bound, which does not give us any algorithmic solution yet. The question we investigate in the next section is

*Can we design an algorithm handling heavy-tailed noise with complexity $\mathcal{O}\left(\frac{\Delta\sigma_h}{\varepsilon^2}\left(\frac{\sigma}{\varepsilon}\right)^{\frac{1}{p-1}}\right)$?*

## 4 Near-optimal Second-Order Method under $p$-BCM

In the last decade, the second-order stochastic optimization (SOSO) has been extensively studied under stronger noise assumptions. We will first review the existing approaches, particularly focusing on the development in the non-convex setup. First, Tripuraneni et al. [2018] proposed and analyzed a Stochastic Cubic Newton (SCN) method, achieving $\bar{\mathcal{O}}(\varepsilon^{-7/2})$ sample complexity. They require a strong bounded noise assumption, use large mini-batches and importantly, this complexity is not order-optimal. Later Arjevani et al. [2020] established lower bounds for SOSO for finding first-order stationary points and proposed two algorithms: SGD with recursive variance reduction and stochastic Hessian-vector products (HVP-RVR), and subsampled cubic-regularized trust-region method with HVP-RVR. Both algorithms achieve $\bar{\mathcal{O}}(\varepsilon^{-3})$ sample complexities, which is minimax optimal when the variances of stochastic gradient and Hessian are bounded. Besides strong noise assumptions, their algorithms require large Hessian batch sizes, since they used HVP-RVR subroutine to update the momentum term. To overcome the large batch-size requirement, Tran and Cutkosky [2021] designed a simpler algorithm: SGD with Hessian-corrected momentum (with optional normalization), called SGDHess. Thanks to the Hessian corrected momentum term, their methods also achieve $\bar{\mathcal{O}}(\varepsilon^{-3})$ sample complexity with any batch size $\geq 1$. Recently, Chayti et al. [2024] proposed a variant of SCN

with momentum without a large batch requirement. However, their analysis requires the bounded noise assumption and only achieves $\bar{\mathcal{O}}(\varepsilon^{-7/2})$ sample complexity.[4]

To summarize, all above-mentioned works require strong noise assumptions: bounded noise and bounded variance. Moreover, their sample complexities also depend on second-order smoothness $L_h$, which can be potentially infinite, as discussed in the previous section. We refer to Table 1 for the summary of existing results.

---

**Algorithm 1** NSGDHess (Normalized SGD with Hessian correction)

---

1: **Input:** Starting point $x_0 \in \mathbb{R}^d$, a vector $g_0 \in \mathbb{R}^d$, a stepsize $\gamma > 0$, momentum parameters $\alpha > 0$, the number of iterations $T$.
2: $x_1 = x_0 - \gamma \frac{g_0}{\|g_0\|}$
3: **for** $t = 1, 2, \ldots, T - 1$ **do**
4:      Sample $q_t \sim \mathcal{U}([0, 1])$
5:      $\hat{x}_t = q_t x_t + (1 - q_t) x_{t-1}$
6:      Sample $\xi_t, \hat{\xi}_t \sim \mathcal{D}$ independently
7:      $g_t = (1 - \alpha) \left( g_{t-1} + \nabla^2 f(\hat{x}_t, \hat{\xi}_t)(x_t - x_{t-1}) \right) + \alpha \nabla f(x_t, \xi_t)$
8:      $x_{t+1} = x_t - \gamma \frac{g_t}{\|g_t\|}$
9: **end for**
10: **Output:**

---

To design an optimal second-order algorithm without these limitations, we take an inspiration from the recent developments in reinforcement learning [Salehkaleybar et al., 2022, Fatkhullin et al., 2023]. Their algorithms called SHARP and (N)HAR-PG are variants of SGDHess in [Tran and Cutkosky, 2021] with optional normalization step. We adapt one variant of their method to our general stochastic optimization setting and present it in Algorithm 1. This algorithm computes a random interpolation point $\hat{x}_t$ uniformly distributed between consecutive iterates $x_{t-1}$ and $x_t$. The method then evaluates a stochastic gradient and a stochastic Hessian-vector product using independent samples $\xi_t$ and $\hat{\xi}_t$. Finally, a recursive momentum is constructed before applying the normalization step.[5] We are now ready to state the convergence guarantee for Algorithm 1.

**Theorem 2.** *Suppose Assumptions 1, 2, 3 and 4 hold. Let the initial gradient estimate be given by*

$$g_0 = \frac{1}{B_{init}} \sum_{j=1}^{B_{init}} \nabla f(x_0, \xi_{0,j}), \quad where \quad B_{init} = \max\left\{ 1, \left(\frac{\sigma}{\varepsilon}\right)^{\frac{p}{p-1}}, \left(\frac{\sigma}{\varepsilon}\right)^{\frac{p}{2p-1}} \right\}.$$

*Set stepsize as* $\gamma = \sqrt{\frac{\Delta \alpha^{1/p}}{(L + \sigma_h) T}}$, *momentum parameter as* $\alpha = \min\{1, \alpha_{\mathit{eff}}\}$, *where* $\alpha_{\mathit{eff}} = \max\left\{ \left(\frac{\mathcal{E}_0}{T\sigma}\right)^{\frac{p}{2p-1}}, \left(\frac{\Delta(L+\sigma_h)}{T\sigma^2}\right)^{\frac{p}{2p-1}} \right\}$, $\mathcal{E}_0 = 2\sigma / B_{init}^{\frac{p-1}{p}}$. *Then, Algorithm 1 guarantees that* $\frac{1}{T} \sum_{t=0}^{T-1} \mathbb{E}\left[\|\nabla F(x_t)\|\right] \leq \varepsilon$ *with total sample complexity*

$$\mathcal{O}\left( \frac{\Delta(L + \sigma_h)}{\varepsilon^2} + \frac{\Delta(L + \sigma_h)}{\varepsilon^2} \left(\frac{\sigma}{\varepsilon}\right)^{\frac{1}{p-1}} + \frac{\sigma}{\varepsilon} \left(\frac{\sigma}{\varepsilon}\right)^{\frac{1}{p-1}} \right). \tag{4}$$

The proof is deferred to Appendix E. We also investigate different choices of $g_0$ and their influence on the total sample complexity rate in Appendix E. We find that while the initial batch size to estimate $g_0$ is not necessary for convergence, it helps to slightly improve the total sample complexity.

**Discussion:** Comparing this result with Theorem 1, we observe that our upper bound (4) matches our lower bound–the first term in (3) in terms of the target accuracy $\varepsilon$. Moreover, when $\Delta(L + \sigma_h) \geq \sigma\varepsilon$, we have a tight upper bound, which exactly matches the lower bound from the previous section in

---

[4]However, in the noiseless case their complexity has better $\bar{\mathcal{O}}(\varepsilon^{-3/2})$ iteration complexity compared to $\bar{\mathcal{O}}(\varepsilon^{-2})$ for SGDHess.

[5]Normalization step for this method is important for our analysis under $p$-BCM. However, it can be removed when $p = 2$, or when additional clipping is used, as in the next section.

the leading term (up to a numerical constant). Moreover, since our upper bound does not depend on constant $L_h$, it explains our observations in Section 3 regarding the limit case $L_h \to \infty$, and answers the raised question affirmatively. We refer to Table 1 for more technical comparison with prior work.

# 5 Hessian Clipping for High-probability Convergence

In Section 4 we provided an in-expectation guarantee for Algorithm 1. While in-expectation guarantees are useful in the case when we are allowed to run a method multiple times to analyze its average-case behavior, it can be impractical. In may real-world scenarios, only a single run of a methods can be performed due to computational or time constraints. Thus, our main goal in this section is to conduct high-probability analysis for Algorithm 1 or its modification.

Since we work in the unbounded variance case, which corresponds to heavy-tailed noise, it is important to ensure that the analyzed method is robust to such noise. Following works of Gorbunov et al. [2020], Cutkosky and Mehta [2021], Sadiev et al. [2023], Nguyen et al. [2023], we propose a new algorithm, called Normalized SGD with momentum and Hessian clipping. It is formally presented as Algorithm 2, where compared to Algorithm 1 we also incorporate gradient clipping and Hessian clipping. Formally, a clipping operator (clipping for short) is defined as

$$\texttt{clip}(v, \lambda) = \min\left\{1, \frac{\lambda}{\|v\|}\right\} v \quad \text{for any } v \neq 0 \text{ from } \mathbb{R}^d, \tag{5}$$

where $\lambda > 0$ is called the clipping level/threshold.

---

**Algorithm 2** Clip NSGDHess (Normalized SGD with Hessian correction and clipping)

1: **Input:** Starting point $x_0 \in \mathbb{R}^d$, a vector $g_0 \in \mathbb{R}^d$, a stepsize $\gamma > 0$, momentum parameters $\alpha > 0$, clipping levels $\lambda > 0, \bar{\lambda}_h > 0$, the number of iterations $T$.
2: $x_1 = x_0$ and $g_0 = 0$
3: **for** $t = 1, 2, \ldots, T - 1$ **do**
4:     Sample $q_t \sim \mathcal{U}([0, 1])$
5:     $\hat{x}_t = q_t x_t + (1 - q_t) x_{t-1}$
6:     Sample $\xi_t, \hat{\xi}_t \sim \mathcal{D}$ independently
7:     $g_t = (1 - \alpha)\left(g_{t-1} + \gamma\, \texttt{clip}\left(\gamma^{-1}\nabla^2 f(\hat{x}_t, \hat{\xi}_t)(x_t - x_{t-1}), \bar{\lambda}_h\right)\right) + \alpha\, \texttt{clip}\left(\nabla f(x_t, \xi_t), \lambda\right)$
8:     $x_{t+1} = x_t - \gamma \frac{g_t}{\|g_t\|}$
9: **end for**
10: **Output:**

---

As in Algorithm 1, we use normalization in the gradient update (line 8), which allows the next point $x_{t+1}$ to stay in the ball with center at $x_t$ of radius $\gamma$. The main difference between Algorithms 1 and 2 is in the momentum term. To enhance robustness, we do not only clip the gradient, but also the Hessian-vector product. As we mentioned above, gradient clipping is a common approach, but Hessian clipping is new, and it allows us to provide the first high-probability guarantees for second-order stochastic optimization without light-tail noise assumptions. It is worth to draw attention to the term

$$\gamma\, \texttt{clip}\left(\gamma^{-1}\nabla^2 f(\hat{x}_t, \hat{\xi}_t)(x_t - x_{t-1}), \bar{\lambda}_h\right). \tag{6}$$

Observe that clipping the entire Hessian based on its operator norm is very costly since it typically requires $\mathcal{O}(d^2)$ arithmetic operations (e.g., using power iteration [Golub and Van Loan, 2013] or Lanczos algorithm [Lanczos, 1950, Saad, 2011]). Instead of clipping the stochastic Hessian, we clip the Hessian-vector product, which is computationally tractable, and only requires $\mathcal{O}(d)$ operations. Thus, the computation of this clipped Hessian-vector product can be easily implemented via backpropagation and subsequent "vector" clipping (5). Another reason why we prefer to use this

form of clipping comes from the analysis; we want to preserve the following useful property:

$$
\begin{aligned}
\mathbb{E}_{q_t,\hat{\xi}_t}\left[\nabla^2 f(\hat{x}_t,\hat{\xi}_t)(x_t-x_{t-1})\right] &= \mathbb{E}_{q_t}\left[\mathbb{E}_{\hat{\xi}_t}\left[\nabla^2 f(\hat{x}_t,\hat{\xi}_t)\right](x_t-x_{t-1})\right] \\
&= \mathbb{E}_{q_t}\left[\nabla^2 F(\hat{x}_t)(x_t-x_{t-1})\right] \\
&= \int_0^1 \nabla^2 F(qx_t+(1-q)x_{t-1})(x_t-x_{t-1})dq \\
&= \nabla F(x_t) - \nabla F(x_{t-1}).
\end{aligned}
\tag{7}
$$

This would not hold were we to use direct Hessian matrix clipping. Instead, we prove that the expectation of (6) will approximately be equal to (7) when $\bar{\lambda}_h$ is selected properly. By smoothness and gradient step, we have $\|\nabla F(x_t) - \nabla F(x_{t-1})\| \leq L\|x_t - x_{t-1}\| = L\gamma$, meaning the expectation is bounded. To provide the analysis, we need to rewrite the clipped Hessian-vector product as follows

$$
\gamma\,\texttt{clip}\left(\gamma^{-1}\nabla^2 f(\hat{x}_t,\hat{\xi}_t)(x_t-x_{t-1}),\bar{\lambda}_h\right) = \texttt{clip}\left(\nabla^2 f(\hat{x}_t,\hat{\xi}_t)(x_t-x_{t-1}),\gamma\bar{\lambda}_h\right),
$$

where we denote $\lambda_h = \gamma\bar{\lambda}_h$. The former formulation in the above formula is useful for implementation, since as we will see, $\lambda_h \sim \mathcal{O}(\gamma\alpha^{-1/p})$ and $\lambda \sim \mathcal{O}(\alpha^{-1/p})$. Thus, $\lambda$ and $\bar{\lambda}_h$ are of the same order $\sim \mathcal{O}(\alpha^{-1/p}) = \mathcal{O}(T^{\frac{1}{2p-1}})$, which makes it easier to tune these two parameters together in practice. The latter formulation will be used in the high-probability analysis. We are now ready to state the main theorem of this section.

**Theorem 3.** *Let Assumptions 1,2,3, and 4 hold, and define the initial optimality gap as $\Delta_1 := F(x_0) - F_*$. Let $\delta \in (0,1]$ and $T \geq 1$ be such that $\log(8T/\delta) \geq 1$. Suppose Algorithm 2 is executed with the following settings: momentum parameter $\alpha = T^{-\frac{p}{2p-1}}$, clipping levels $\lambda = \max\left\{4\sqrt{L\Delta_1}, \frac{\sigma}{\alpha^{1/p}}\right\}$, $\lambda_h = \frac{2\gamma(L+\sigma_h)}{\alpha^{1/p}}$, stepsize $\gamma$ satisfying:*

$$
\gamma \leq \mathcal{O}\left(\min\left\{\sqrt{\frac{\Delta_1}{LT}}, \alpha\sqrt{\frac{\Delta_1}{L}}, \frac{\sqrt{\Delta_1/L}}{\alpha T\log(T/\delta)}, \frac{\Delta_1}{\sigma\alpha^{\frac{p-1}{p}}T\log(T/\delta)}, \sqrt{\frac{\Delta_1\alpha^{1/p}}{(L+\sigma_h)T\log(T/\delta)}}\right\}\right).
$$

*Then, with probability at least $1-\delta$, the output of the algorithm satisfies: $\frac{1}{T}\sum_{t=0}^{T-1}\|\nabla F(x_t)\| \leq \frac{2\Delta_1}{\gamma T}$. As a result, the gradient norm converges at the rate*

$$
\frac{1}{T}\sum_{t=0}^{T-1}\|\nabla F(x_t)\| = \mathcal{O}\left(\frac{\sqrt{\Delta_1(L+\sigma_h)}+\sigma}{T^{\frac{p-1}{2p-1}}}\log\left(\frac{T}{\delta}\right)\right)
$$

*with high probability.*

The complete proof of Theorem 3 can be found in Appendix F. Disregarding the parameters $L, \sigma, \sigma_h, \Delta_1$ in the choices of the stepsize and the clipping level, we see that the momentum parameter and the stepsize decrease with the same rate $\sim \mathcal{O}(1/T^{\frac{p}{p-1}})$, similarly to Algorithm 1. Also, as we mentioned above, clipping levels $\lambda$ and $\bar{\lambda}_h$ have the same rate, and behave as $\sim \mathcal{O}(T^{\frac{1}{2p-1}})$.

**Corollary 1.** *In the setting of Theorem 3, Algorithm 2 ensures that $\frac{1}{T}\sum_{t=0}^{T-1}\|\nabla F(x_t)\| \leq \varepsilon$, with probability at least $1-\delta$. To achieve this, the algorithm requires at most*

$$
\mathcal{O}\left(\left(\frac{\sqrt{\Delta_1(L+\sigma_h)}+\sigma}{\varepsilon}\log\frac{1}{\delta\varepsilon}\right)^{\frac{2p-1}{p-1}}\right) \quad \textit{first/second-order oracle calls.}
$$

According to the results of Theorem 3 and Corollary 1, we have shown that Algorithm 2 has a near-optimal convergence rate in the dependence on $\varepsilon$. Moreover, the complexity has a logarithmic dependence on $1/\delta$, which is better than a polynomial dependence that can be achieved by directly translating the in-expectation result of Theorem 2 to a high probability one using Markov's inequality. Unfortunately, the dependence on $\Delta_1, L, \sigma, \sigma_h$ does not match the lower bound (3).

**Comparison with [Liu et al., 2023b].** Recently, and under additional strong assumptions, Liu et al. [2023b] analyzed a momentum-based, variance-reduced first-order method, and obtained rates

similar to those in our Corollary 1. However, they assume individual smoothness (i.e., $\|\nabla f(x, \xi) - \nabla f(y, \xi)\| \le \ell \|x - y\|$ for all $x, y \in \mathbb{R}^d$ and all realizations of the random variable $\xi$). However, individual smoothness is a very strong assumption which implies boundedness of the stochastic Hessian, i.e., $\|\nabla^2 f(x, \xi)\| \le \ell$ a.s., for all $x \in \mathbb{R}^d$. We also note that under individual smoothness, fast $p$-independent rates can be obtained, e.g., Lei et al. [2019] achieve $\bar{\mathcal{O}}(\varepsilon^{-4})$ complexity for SGD under individual smoothness.

## 6 Limitations and Future Work

While our proposed method with Hessian clipping has many desirable properties, such as near-optimal sample complexity, batch-free and high-probability convergence, under mild statistical assumptions, it also has several limitations. First, it requires tuning three extra parameters compared to, e.g., vanilla SGD [Fatkhullin et al., 2025] or Normalized SGD [Hübler et al., 2025]. It would be interesting to investigate if such extensive tuning can be removed for second-order methods. Second, while our sample complexity is near optimal, it does not match our lower bound exactly in all important parameters. Moreover, while we mainly pay attention to the leading stochastic terms in the complexity, our methods do not recover optimal deterministic complexities of second-order methods. Perhaps a different analysis technique for clipping and some algorithmic modifications (like Newton type preconditioning [Chayti et al., 2024]) are required to exactly match the upper and lower bounds. Third, high probability upper bounds are derived for a fixed confidence level $\delta$ rather than uniformly for all $\delta \in (0, 1)$ (as e.g., in [Liu et al., 2023a, Hübler et al., 2025]) and the algorithm's parameters depend on this choice. This is limiting as it does not necessarily imply high probability convergence for a fixed parameter choice. Finally, while we make an important progress in the fundamental understanding of second-order stochastic optimization, such results should always be interpreted with caution. In practice, the overhead of second-order methods (computing Hessian information or even Hessian-vector products) and the challenge of tuning additional hyperparameters (such as clipping thresholds for both gradients and Hessians) can be significant.

## Acknowledgements

The research reported in this publication was supported by funding from King Abdullah University of Science and Technology (KAUST): i) KAUST Baseline Research Scheme, ii) CRG Grant ORFS-CRG12-2024-6460, and iii) Center of Excellence for Generative AI, under award number 5940. The work of I.F. is supported by ETH AI Center Doctoral Fellowship.

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

# Contents

Table 1: Summary of sample complexities of stochastic second-order (SOSO) methods for finding an $\varepsilon$-stationary point, in high probability of in expectation, i.e., number of stochastic gradient and Hessian evaluations to find $\bar{x}$ with $\|\nabla F(\bar{x})\| \leq \varepsilon$. The column "$p$" indicates the range of moments in Assumptions 3, 4 for which the result holds, that is when $p = 2$, the corresponding result holds only under bounded variance. When $p = \infty$ it means that at least for stochastic gradient or stochastic Hessian bounded noise assumption is required, which corresponds to all moments being bounded. We are not aware of any prior works for SOSO for $p < 2$. The column "**HP?**" denotes whether the high probability guarantee with polylogarithmic dependence on the inverse of failure probability $1/\delta$ is available.

| Algorithm | Sample Complexity | $p$ | HP? |
|---|---|---|---|
| SCN [Tripuraneni et al., 2018] | $\frac{\Delta\sqrt{L_h}\sigma^2}{\varepsilon^{7/2}}$ | $\infty$ | ✓[1] |
| SGD with HVP-RVR [Arjevani et al., 2020] | $\frac{\Delta\sigma\sigma_h}{\varepsilon^3} + \frac{\Delta\sqrt{L_h}\sigma_1}{\varepsilon^{5/2}} + \frac{\Delta L}{\varepsilon^2}$ | $2$ | ✗ |
| SN[2] with HVP-RVR [Arjevani et al., 2020] | $\frac{\Delta\sigma\sigma_h}{\varepsilon^3} + \frac{\Delta\sqrt{L_h}\sigma_1}{\varepsilon^{5/2}} + \frac{\Delta\sigma_h}{\varepsilon^2}$ | $2$ | ✗ |
| N-SGDHess [Tran and Cutkosky, 2021] | $\frac{\sigma^3}{\varepsilon^3} + \frac{\Delta\sigma\sigma_h}{\varepsilon^3} + \frac{\Delta\sqrt{L_h}\sigma_1}{\varepsilon^{5/2}} + \frac{\Delta\sigma_h}{\varepsilon^2}$ | $2$ | ✗ |
| SCN with IT-HB [Chayti et al., 2024] | $\frac{\Delta\sqrt{L_h}}{\varepsilon^{3/2}} + \frac{\Delta L_h^{1/4}\sigma_h^{1/2}}{\varepsilon^{7/4}} + \frac{\Delta\sqrt{L}\sigma^2}{\varepsilon^{7/2}}$ | $\infty$ | ✗ |
| Lower Bounds Theorem 1 | $\min\left\{\frac{\Delta\sigma_h}{\varepsilon^2}, \frac{\Delta L\sigma}{\varepsilon^3}, \frac{\Delta\sqrt{L_h}\sigma}{\varepsilon^{5/2}}\right\}\left(\frac{\sigma}{\varepsilon}\right)^{\frac{1}{p-1}}$ | $(1,2]$ | |
| NSGDMHess Theorem 2 | $\frac{\Delta(L+\sigma_h)}{\varepsilon^2} + \left(\frac{\Delta(L+\sigma_h)}{\varepsilon^2} + \frac{\sigma}{\varepsilon}\right)\left(\frac{\sigma}{\varepsilon}\right)^{\frac{1}{p-1}}$ | $(1,2]$ | ✗ |
| Clip NSGDMHess Theorem 3 | $\left(\frac{\sqrt{\Delta_1(L+\sigma_h)}+\sigma}{\varepsilon}\right)^{2+\frac{1}{p-1}}$ | $(1,2]$ | ✓ |

[1] Tripuraneni et al. [2018] provide analysis under stronger assumptions, which implies the noise has light tails.
[2] SN = Subsampled Newton.

# A  Additional Related Work

**Gradient clipping and normalization.**  Gradient clipping has been applied successfully also in zero-order optimization [Kornilov et al., 2024], bandit and RL literature [Bubeck et al., 2013, Cayci and Eryilmaz, 2024], online learning [Zhang and Cutkosky, 2022] and differential privacy [Abadi et al., 2016, Sha et al., 2024]. Some other works, which use normalization and gradient clipping under heavy-tailed noise include [Armacki et al., 2023, Puchkin et al., 2024, Liu and Zhou, 2024].

# B  About Normalized SGD and Different Variants of Momentum

In this section, we present a general framework for momentum-based methods. The study of momentum has a long history, particularly in convex optimization, where Nesterov [1983] demonstrated that an appropriate use of momentum in gradient descent can accelerate convergence. In contrast, our focus is on momentum in the setting of stochastic non-convex optimization. The general update rule is given by

$$x_{t+1} = x_t - \gamma_t \frac{g_t}{\|g_t\|},$$

where $g_t$ denotes the momentum term. Different choices of $g_t$ give rise to different algorithms, several of which we now describe.

**NSGD** is the standard stochastic gradient method. Setting $g_t = \nabla f(x_t, \xi_t)$ in the general framework immediately recovers NSGD. It is well known that NSGD converges only to a neighborhood of the solution, with radius proportional to $\sigma$ [Yang et al., 2023, Hübler et al., 2025]. One possible remedy is to employ mini-batching, which reduces the variance. However, this strategy cannot be applied if only a single sample per iteration is allowed.

**NSGDM** is the solution of the problem described above and was proposed and analyzed by Cutkosky and Mehta [2020]. The momentum term is defined as

$$g_t = (1 - \alpha)g_{t-1} + \alpha\nabla f(x_t, \xi_t).$$

In their work, the authors proved that NSGDM converges to an $\varepsilon$-stationary point and achieves a sample complexity of $\mathcal{O}(\varepsilon^{-4})$, which is optimal according to the lower bound of Arjevani et al. [2023]. Subsequently, Hübler et al. [2025] extended this result to the setting where the stochastic gradients have bounded $p$-th moments, deriving a sample complexity of $\mathcal{O}(\varepsilon^{-\frac{3p-2}{p-1}})$.

**NSGDM with clipping** was proposed for the general setting where the stochastic gradients have bounded $p$-th moments. The momentum term is modified as

$$g_t = (1 - \alpha)g_{t-1} + \alpha, \texttt{clip}\left(\nabla f(x_t, \xi_t), \lambda\right).$$

Cutkosky and Mehta [2021] established high-probability convergence guarantees and showed that this method achieves a sample complexity of $\mathcal{O}(\varepsilon^{-\frac{3p-2}{p-1}})$, which is optimal and matches the lower bounds of Zhang et al. [2020]. It is worth noting that the clipping operator is not strictly necessary: normalization of the momentum term is sufficient to guarantee convergence under Assumption 3 [Hübler et al., 2025, Liu and Zhou, 2024].

**NIGT** is a variant of NSGD that incorporates momentum and implicit gradient transport, proposed by Cutkosky and Mehta [2020]. In this method, the momentum term is modified by changing the point at which the stochastic gradient is evaluated:

$$g_t = (1 - \alpha)g_{t-1} + \alpha\nabla f(y_t, \xi_t), \quad \text{where } y_t = x_t + \frac{1 - \alpha}{\alpha}x_{t-1}.$$

Thanks to this modification, and under the assumption that the objective function has a Lipschitz continuous Hessian, Cutkosky and Mehta [2020] proved that NIGT achieves a sample complexity of $\mathcal{O}\left(\varepsilon^{-7/2}\right)$, which improves upon that of NSGDM.

**NIGT with clipping** was introduced by Cutkosky and Mehta [2021] to enable high-probability analysis under the assumption of bounded $p$-th moments of the stochastic gradients. The only modification compared to standard NIGT is the inclusion of a clipping operator, which is required from a theoretical standpoint. As shown in Cutkosky and Mehta [2021], NIGT with clipping achieves a sample complexity of $\mathcal{O}\left(\varepsilon^{-\frac{5p-3}{2p-2}}\right)$, which is optimal according to Theorem 1.

**NSGD with MVR** is a variant of NSGD that incorporates momentum variance reduction (MVR). The STORM/MVR method was proposed by Cutkosky and Orabona [2019] to improve the sample complexity compared to standard SGD. More recently, He et al. [2025] established that NSGD with MVR achieves a sample complexity of $\mathcal{O}\left(\varepsilon^{-\frac{2p-1}{p-1}}\right)$, which matches ours in terms of dependence on $\varepsilon$ (Theorem 2).

The methods discussed above rely solely on first-order information. A natural question is whether incorporating second-order information can further improve the sample complexity.

**NSGD-Hess** is a variant of Normalized SGD that incorporates Hessian-corrected momentum, proposed by Tran and Cutkosky [2021]. In this method, a stochastic Hessian-vector product is introduced into the momentum update:

$$g_t = (1 - \alpha)\left(g_{t-1} + \nabla^2 f(x_t, \xi_t) \cdot (x_t - x_{t-1})\right) + \alpha\nabla f(y_t, \xi_t).$$

**Refined NSGD-Hess.** In our refined version, the momentum term is modified by adjusting the point at which the stochastic Hessian is evaluated and by using an independent fresh sample $\hat{\xi}_t \sim \mathcal{D}$ for the Hessian-vector product computation:

$$
\begin{aligned}
\hat{x}_t &= q_t x_t + (1 - q_t)x_{t-1}, \quad \text{where } q_t \sim \mathcal{U}([0, 1]), \\
g_t &= (1 - \alpha)\left(g_{t-1} + \nabla^2 f(\hat{x}_t, \hat{\xi}_t) \cdot (x_t - x_{t-1})\right) + \alpha\nabla f(x_t, \xi_t).
\end{aligned}
$$

## C Technical Lemmas

We list the following technical lemmas, which are useful for our analysis in the subsequent sections.

**Lemma 1** (Lemma 10 from Hübler et al. [2025]). *Let $p \in [1; 2]$, and $X_1, \ldots, X_n \in \mathbb{R}^d$ be a martingale difference sequence, i.e. $\mathbb{E}[X_j | X_{j-1}, \ldots, X_1] = 0$ a.s. for all $j = 1, \ldots, n$ satisfying*

$$\mathbb{E}[\|X_j\|^p] < \infty \qquad \text{for all } i = 1, \ldots, n.$$

*Define $S_n \stackrel{def}{=} \sum_{j=1}^n X_j$, then*

$$\mathbb{E}[\|S_n\|^p] \le 2 \sum_{i=j}^n \mathbb{E}[\|X_j\|^p].$$

**Lemma 2** (Lemma 10 from Cutkosky and Mehta [2021]). *Let $X_1, \ldots, X_n \in \mathbb{R}^d$ be a sequence of random vectors. Define the sequence of real numbers $w_1, \ldots, w_n$ recursively*

1. $w_0 = 0$

2. *If $\sum_{i=1}^{j-1} X_i \neq 0$, then we set:*

$$w_j = \text{sign}\left(\sum_{i=1}^{j-1} w_i\right) \frac{\left\langle \sum_{i=1}^{j-1} X_i, X_j \right\rangle}{\left\| \sum_{i=1}^{j-1} X_i \right\|}.$$

3. *If $\sum_{i=1}^{j-1} X_i = 0$, set $w_j = 0$.*

*Then $|w_j| \le \|X_j\|$ for all $j = 1, \ldots, n$, and*

$$\|S_n\| \le \left| \sum_{j=1}^n w_j \right| + \sqrt{\max_{j \in [n]} \|X_j\|^2 + \sum_{j=1}^n \|X_j\|^2}. \tag{8}$$

**Lemma 3** (Bernstein inequality). *Let $X_1, \ldots, X_n \in \mathbb{R}^d$ be a martingale difference sequence, i.e. $\mathbb{E}[X_j | X_{j-1}, \ldots, X_1] = 0$ a.s. for all $j = 1, \ldots, n$. Assume that conditional variances $\sigma_j^2 \stackrel{def}{=} \mathbb{E}[X_j^2 | X_{j-1}, \ldots, X_1]$ exist and are bounded, and assume that there exists deterministic constant $c > 0$ such that $|X_j| \le c$ almost surely for all $j = 1, \ldots, n$. Define $S_n \stackrel{def}{=} \sum_{i=1}^n X_j$, then for all $b > 0, G > 0$*

$$\mathbb{P}\left\{ |S_n| > b \text{ and } \sum_{j=1}^n \sigma_i^2 \le G \right\} \le 2 \exp\left( -\frac{b^2}{2G + 2cb/3} \right), \tag{9}$$

**Lemma 4** ( Lemma 5.1 from Sadiev et al. [2023]). *Let $\lambda > 0$ and $X \in \mathbb{R}^d$ be a random vector and $\widetilde{X} \stackrel{def}{=} \texttt{clip}(X, \lambda)$. Then, $\left\| \widetilde{X} - \mathbb{E}[\widetilde{X}] \right\| \le 2\lambda$. Moreover, if for some $\sigma \ge 0$ and $p \in (1; 2]$ we have $\mathbb{E}[X] = x \in \mathbb{R}^d$, $\mathbb{E}[\|X - x\|^p] \le \sigma^p$, and $\|x\| \le \lambda/2$, then*

$$\left\| \mathbb{E}[\widetilde{X}] - x \right\| \le \frac{2^p \sigma^p}{\lambda^{p-1}}, \quad \mathbb{E}\left[ \left\| \widetilde{X} - x \right\|^2 \right] \le 18\lambda^{2-p}\sigma^p, \quad \mathbb{E}\left[ \left\| \widetilde{X} - \mathbb{E}[\widetilde{X}] \right\|^2 \right] \le 18\lambda^{2-p}\sigma^p.$$

# D    Missing Proofs for Section 3

In this section we provide lower bounds for the certain class of functions and the oracle class. Our prove is inspired by the paper of Arjevani et al. [2020].

**Auxiliary Lemmas.** Now we state two helpful lemmas to prove lower bound. The first lemma is about the number of iteration to reveal all coordinates, when zero-respecting algorithms use the information received from the oracle forming *probability-$\rho$ zero-chain*.

**Definition 2.** *A collection of derivative estimators $\nabla^1 f(x,\xi)$, $\nabla^2 f(x,\xi)$, ..., $\nabla^q f(x,\xi)$ for a function $F$ forms a probability-$\rho$ zero-chain if*

$$\mathbb{P}\left\{\exists\, x \mid prog\left(\nabla^1 f(x,\xi),\ldots,\nabla^q f(x,\xi)\right) = prog_{\frac{1}{4}}(x) + 1\right\} \leq \rho$$

*and*

$$\mathbb{P}\left\{\exists\, x \mid prog\left(\nabla^1 f(x,\xi),\ldots,\nabla^q f(x,\xi)\right) = prog_{\frac{1}{4}}(x) + i\right\} = 0, \; i > 1.$$

The second lemma is about the main properties of the *worst* function from the class $\mathcal{F}(\Delta, L_{1:q})$.

**Lemma 5** ([Arjevani et al., 2020]). *Let $\nabla^1 f(x,\xi),\ldots,\nabla^q f(x,\xi)$ be a collection of probability-$\rho$ zero-chain derivative estimators for $F : \mathbb{R}^T \to \mathbb{R}$, and let $O_F^p$ be an oracle with $O_f^p = (\nabla^r f(x,\xi))_{r\in[q]}$. Let $\{x_{A[O_F^q]}^{(t)}\}$ be a sequence of queries produced by a zero-respecting algorithm $A$ interacting with $O_F^q$. Then, with probability at least $1-\delta$*

$$prog\left(x_{A[O_F^q]}^{(t)}\right) < T, \qquad for\ all \quad t \leq \frac{T - \log\frac{1}{\delta}}{2\rho}.$$

**Lemma 6** (Carmon et al. [2020]). *Let $h : \mathbb{R}^T \to \mathbb{R}$ be the following function:*

$$h(x) = -\Psi(1)\Phi(x_1) + \sum_{i=2}^{T}\left(\Psi(-x_{i-1})\Phi(-x_i) - \Psi(x_{i-1})\Phi(x_i)\right),$$

*where*

$$\Psi(x) = \begin{cases} 0, & x \leq \frac{1}{2}; \\ \exp\left(1 - \frac{1}{(2x-1)^2}\right), & x > \frac{1}{2}, \end{cases} \quad and \quad \Phi(x) = \sqrt{e}\int_{-\infty}^{x} e^{-\frac{1}{2}t^2}\, dt.$$

*Then the function $h$ satisfies the following properties:*

- $h(0) - \inf_x h(x) \leq \Delta_0 T$, *where* $\Delta_0 = 12$.

- *For $q \geq 1$, the $q$-th order derivatives of $h$ are $\ell_q$-Lipschitz continuous, where $\ell_q \leq \exp\left(\frac{5}{2}q\log q + cq\right)$ for a numerical constant $c < \infty$.*

- *For all $x \in \mathbb{R}^T$, $q \in \mathbb{N}$ and $i \in [T]$, we have $\|\nabla_i^q h(x)\|_{op} \leq \ell_{q-1}$, where $\ell_0 = 23$.*

- *For all $x \in \mathbb{R}^T$ and $q \in \mathbb{N}$, $prog\left(\nabla^q h(x)\right) \leq prog_{\frac{1}{2}}(x) + 1$.*

- *For all $x \in \mathbb{R}^T$, if $prog_1(x) < T$, then $\|\nabla h(x)\| \geq |\nabla_{prog_1(x)+1} h(x)| > 1$.*

**Lemma 7.** *Let the derivative estimator is defined as follows for each $r \in [q]$*

$$[\nabla^r h(x,\xi)]_i \overset{def}{=} \left(1 + \mathbb{I}\left\{i > prog_{\frac{1}{4}}(x)\right\}\left(\frac{\xi}{\rho} - 1\right)\right)\cdot \nabla_i^r h(x),$$

*where $\xi \sim Bernoulli(\rho)$. Then, $\{\nabla^r h(x,\xi)\}_{r\in[q]}$ forms probability-$\rho$ zero-chain, and for any $p \in [1,2]$ and for each $r \in [q]$ the derivative estimator satisfies*

$$\mathbb{E}\left[\nabla^r h(x,z)\right] = \nabla^r h(x), \quad and \quad \mathbb{E}\left[\|\nabla^r h(x,\xi) - \nabla^r h(x)\|_{op}^p\right] \leq \frac{2\ell_{r-1}^p}{\rho^{p-1}} \qquad for\ all\ x \in \mathbb{R}^d.$$

*Proof.* First of all, it is easy to show that the proposed estimator is unbiased, i.e.
$$\mathbb{E}\left[\nabla^r h(x,\xi)\right] = (1+0)\cdot\nabla^r h(x) = \nabla^r h(x).$$
According to Lemma 6, we have $\text{prog}\left(\nabla^r h(x)\right) < \text{prog}_{\frac{1}{2}}(x)+1 \le \text{prog}_{\frac{1}{4}}(x)+1$, where the last inequality is true because $\text{prog}_\alpha$ is non-increasing with respect to $\alpha$. Then, by Lemma 6 we have
$$\left[\nabla^r h(x,\xi)\right]_i = \nabla_i^r h(x) = 0, \quad \forall i > \text{prog}_{\frac{1}{4}}(x)+1.$$
Therefore, due to that $\xi \sim \text{Bernoulli}(\rho)$ we obtain
$$\mathbb{P}\left\{\exists x \mid \text{prog}\left(\nabla^1 f(x,z),\ldots,\nabla^q f(x,z)\right) = \text{prog}_{\frac{1}{4}}(x)+1\right\} \le \rho,$$
i.e. $\{\nabla^r h(x,\xi)\}_{r\in[q]}$ forms probability-$\rho$ zero-chain.

Now we bound $p$-th moment of $\nabla^r h(x,\xi) - \nabla^r h(x)$ for any $p \in [1;2]$. Denoting $i_x = \text{prog}_{\frac{1}{4}}(x)+1$, for any $r \in [q]$ we have

$$
\begin{aligned}
\mathbb{E}\left[\|\nabla^r h(x,\xi) - \nabla^r h(x)\|_{\text{op}}^p\right] &= \mathbb{E}\left[\left|\mathbb{I}\left\{i_x > \text{prog}_{\frac{1}{4}}(x)\right\}\left(\frac{\xi}{\rho}-1\right)\right|^p \|\nabla_i^r h(x)\|_{\text{op}}^p\right] \\
&= \mathbb{E}\left[\left|\frac{\xi}{\rho}-1\right|^p\right]\|\nabla_i^r h(x)\|_{\text{op}}^p \\
&\le \ell_{r-1}^p \mathbb{E}\left[\left|\frac{\xi}{\rho}-1\right|^p\right] \\
&= \ell_{r-1}^p \frac{(1-\rho)^p\rho + (1-\rho)\rho^p}{\rho^p} \\
&\le \ell_{r-1}^p \frac{2}{\rho^{p-1}}.
\end{aligned}
$$

$\square$

**Main Theorem.** Now we are ready to state and prove the main theorem in this section, i.e. Theorem 1.

*Proof.* First of all, we fix all parameters: $q \in \mathbb{N}, \Delta, L_{1:q}, \sigma_{1:q} > 0$ and $\varepsilon > 0$. Next, we rescale our function $h$ as follows: $h^\star(x) = \nu h(\beta x)$, where $\nu > 0$ and $\beta > 0$. According to Lemma 6, selecting $T = \left\lfloor \frac{\Delta}{\nu\Delta_0} \right\rfloor$, we have for any $r \in [q-1]$

$$
\begin{aligned}
h^\star(0) - \inf_x h^\star(x) &= \nu\left(h^\star(0) - \inf_x h^\star(\beta x)\right) \le \nu\Delta_0 T \le \Delta; \\
\|\nabla^{r+1}h^\star(x)\|_{\text{op}} &= \nu\beta^{r+1}\|\nabla^{r+1}h(\beta x)\| \le \nu\beta^{r+1}\ell_r; \\
\|\nabla h^\star(x)\| &= \nu\beta\|\nabla h(\beta x)\| \ge \nu\beta\|\nabla h(x)\| \ge \nu\beta, \ \forall\, x: \ \text{prog}_1(x) < T.
\end{aligned}
$$

Since $\left\{\nabla^r h^\star(x,\xi) = \nu\beta^r\nabla^r h(\beta x,\xi)\right\}_{r\in[p]}$ forms a probability-$\rho$ zero-chain, then by Lemma 6 with probability at least $\delta = \frac{1}{2}$ we have

$$\mathbb{E}\left[\left\|\nabla h^\star(x_{A[O_h^q]}^t)\right\|\right] = \nu\beta\mathbb{E}\left[\left\|\nabla h^\star(\beta x_{A[O_h^q]}^t)\right\|\right] \ge \frac{\nu\beta}{2}, \quad \forall\, t \le \frac{T-1}{2\rho}$$

The $p$-th central moment of the scaled derivatives estimators is bounded as

$$
\begin{aligned}
\mathbb{E}\left[\|\nabla^r h^\star(x,\xi) - \nabla^r h^\star(x)\|_{\text{op}}^p\right] &\le \nu^p\beta^{rp}\mathbb{E}\left[\|\nabla^r h(\beta x,\xi) - \nabla^r h(\beta x)\|_{\text{op}}^p\right] \\
&\le \frac{2\nu^p\beta^{rp}\ell_{r-1}^p}{\rho^{p-1}},
\end{aligned}
$$

where in the last inequality we applied Lemma 7. Thus, we have the set of constraints for parameters $\nu$ and $\beta$ for all $r \in [q]$:

$$
\begin{cases}
\nu\Delta_0 T \le \Delta; \\
\nu\beta^{r+1}\ell_r \le L_r; \\
\frac{\nu\beta}{2} \ge \varepsilon; \\
\nu^p\beta^{pr}\ell_{r-1}^p\frac{2}{\rho^{p-1}} \le \sigma_r^p.
\end{cases}
$$

We will resolve the system of inequalities step by step. The first step is to set $\nu = \frac{2\varepsilon}{\beta}$. For $r = 1$, we have

$$\nu^p \beta^p \ell_0^p \cdot \frac{2}{\rho^{p-1}} \leq \sigma_1^p \quad \Rightarrow \quad \rho = \min\left\{ \left(\frac{\varepsilon}{\sigma_1}\right)^{\frac{-p}{p-1}} \cdot 2^{\frac{p+1}{p-1}} \ell_0^{\frac{-p}{p-1}}, 1 \right\}.$$

To set $\beta$, we need to find its specific value, which can satisfy the following constraints: for any $r \in \{2, \dots, q\}$ and any $r' \in [q]$

$$\beta^{r+1} \leq \frac{L_r}{\nu \ell_r} = \frac{\beta L_r}{2\varepsilon \ell_r}, \quad \nu^p \beta^{pr} \ell_{r-1}^p \frac{2}{\rho^{p-1}} \overset{(*)}{\leq} \beta^{p(r-1)} \left(\frac{\sigma_1 \ell_{r-1}}{\ell_0}\right)^p \leq \sigma_r^p,$$

where in the inequality $(*)$ we used $\frac{2}{\rho^{p-1}} \leq \frac{\sigma_1^p}{\nu^p \beta^p \ell_0^p}$. Therefore, we obtain

$$\beta = \min_{r' \in [q];\; r \in \{2,\dots,q\}} \min\left\{ \left(\frac{\ell_0 \sigma_r}{\ell_{r-1}\sigma_1}\right)^{\frac{1}{r-1}}, \left(\frac{L_{r'}}{2\varepsilon \ell_{r'}}\right)^{\frac{1}{r'}} \right\}.$$

Then, assuming $T \geq 3$ and $\left(\frac{\varepsilon}{\sigma_1}\right)^{\frac{-p}{p-1}} \cdot 2^{\frac{p+1}{p-1}} \ell_0^{\frac{p}{p-1}} \leq 1$, we get

$$\begin{aligned}
\frac{T-1}{2\rho} &= \frac{1}{2\rho}\left(\left\lfloor \frac{\Delta\beta}{2\Delta_0\varepsilon}\right\rfloor - 1\right) \geq \frac{\Delta\beta}{8\rho\Delta_0\varepsilon}\\
&\geq \left(\frac{\sigma_1}{2\varepsilon\ell_0}\right)^{\frac{p}{p-1}} \cdot \frac{\Delta}{4\Delta_0\varepsilon} \min_{r' \in [q];\; r \in \{2,\dots,q\}} \min\left\{ \left(\frac{\ell_0\sigma_r}{\ell_{r-1}\sigma_1}\right)^{\frac{1}{r-1}}, \left(\frac{L_{r'}}{2\varepsilon\ell_{r'}}\right)^{\frac{1}{r'}} \right\}\\
&\geq \Omega(1) \cdot \frac{\Delta}{\varepsilon}\left(\frac{\sigma_1}{\varepsilon}\right)^{\frac{p}{p-1}} \min\left\{ \min_{r \in \{2,\dots,q\}} \left(\frac{\sigma_r}{\sigma_1}\right)^{\frac{1}{r-1}}, \min_{r' \in [q]} \left(\frac{L_{r'}}{\varepsilon}\right)^{\frac{1}{r'}} \right\}.
\end{aligned}$$

Since $h$ and $h^\star$ are functions of $T$ arguments, the dimension of the problem is equal to $T$. This concludes the proof. $\qquad\square$

# E Missing Proofs for Section 4

In this section we study Normalized SGD with Hessian correction (NSGD-Hess) (see Algorithm 1).

**Auxiliary Lemmas.** To show the convergence guarantees for Algorithm 1, we state and prove auxiliary lemmas. The first one is well-known *Descent Lemma* for Normalized SGD. The proof for this lemma can be found in Cutkosky and Mehta [2020], Hübler et al. [2025], but for completeness we restate and reprove it. The second one is devoted to the bounds on *error* term from *Descent Lemma*.

**Lemma 8.** *Let Assumptions 1 and 2 hold. Then for any selection of stepsize $\gamma > 0$ the iterates $\{x_t\}_{t=0}^T$ generated by Algorithm 1 satisfy*

$$\gamma \sum_{t=0}^{T-1} \|\nabla F(x_t)\| + \Delta_T \leq \Delta_0 + 2\gamma \sum_{t=0}^{T-1} \|\hat{e}_t\| + \frac{\gamma^2 LT}{2}. \tag{10}$$

*where functional gap is defined as $\Delta_t \overset{def}{=} F(x_t) - F_*$, error term is defined as $\hat{e}_t \overset{def}{=} g_t - \nabla F(x_t)$.*

*Proof.* According to the update rule for $x_t$ and Assumption 2, we have

$$
\begin{aligned}
F(x_{t+1}) &\leq F(x_t) + \langle \nabla F(x_t), x_{t+1} - x_t \rangle + \frac{L}{2}\|x_{t+1} - x_t\|^2 \\
&= F(x_t) - \gamma \left\langle \nabla F(x_t), \frac{g_t}{\|g_t\|} \right\rangle + \frac{\gamma^2 L}{2} \\
&= F(x_t) - \gamma\|g_t\| - \gamma \left\langle \nabla F(x_t) - g_t, \frac{g_t}{\|g_t\|} \right\rangle + \frac{\gamma^2 L}{2} \\
&\leq F(x_t) - \gamma\|g_t\| + \gamma\|\nabla F(x_t) - g_t\| + \frac{\gamma^2 L}{2} \\
&\leq F(x_t) - \gamma\|\nabla F(x_t)\| + 2\gamma\|\nabla F(x_t) - g_t\| + \frac{\gamma^2 L}{2}.
\end{aligned}
$$

Using notation for the functional gap and the error term, we get

$$\gamma\|\nabla F(x_t)\| + \Delta_{t+1} \leq \Delta_t + 2\gamma\|\hat{e}_t\| + \frac{\gamma^2 L}{2}.$$

Summing over $t$ from 0 to $T - 1$, we obtain

$$\gamma \sum_{t=0}^{T-1} \|\nabla F(x_t)\| + \Delta_T \leq \Delta_0 + 2\gamma \sum_{t=0}^{T-1} \|\hat{e}_t\| + \frac{\gamma^2 LT}{2}. \tag{11}$$

$\square$

Next, we derive the lemma for error control for the Hessian corrected momentum estimator (lines $4-7$ in Algorithm 1). Similar lemma for the special case $p = 2$ appeared previously in [Salehkaleybar et al., 2022, Fatkhullin et al., 2023]. For the case $p < 2$, similar recursion was derived by Hübler et al. [2025] for first-order momentum. Now we extend this idea to the case of second-order momentum.

**Lemma 9.** *Let Assumptions 2, 3 and 4 hold. Then for all $t \geq 0$, we have*

$$\mathbb{E}\left[\|\hat{e}_t\|\right] \leq (1 - \alpha)^t \mathbb{E}\left[\|\hat{e}_0\|\right] + 2\alpha^{\frac{p-1}{p}}\sigma + 12\gamma(L + \sigma_h)\alpha^{-1/p}, \tag{12}$$

*where $\hat{e}_t := g_t - \nabla F(x_t)$.*

*Proof.* Define $e_t = \nabla f(x_t, \xi_t) - \nabla F(x_t)$, $\hat{S}_{t+1} = \nabla^2 f(\hat{x}_{t+1}, \hat{\xi}_{t+1}) \cdot (x_{t+1} - x_t) - \nabla F(x_{t+1}) + \nabla F(x_t)$. We have $\mathbb{E}[e_t] = 0$, $\mathbb{E}[\|e_t\|^p] \leq \sigma^p$, $\mathbb{E}\left[\hat{S}_t\right] = 0$, and

$$
\begin{aligned}
\mathbb{E}\left[\left\|\hat{S}_t\right\|^p\right] &= \mathbb{E}\left[\left\|\nabla F(x_t) - \nabla F(x_{t+1}) + \nabla^2 f(\hat{x}_{t+1}, \hat{\xi}_{t+1}) \cdot (x_{t+1} - x_t)\right\|^p\right] \\
&\leq 3\mathbb{E}\left[\|\nabla F(x_t) - \nabla F(x_{t+1})\|^p\right] + 3\mathbb{E}\left[\left\|\nabla^2 F(\hat{x}_{t+1}) \cdot (x_{t+1} - x_t)\right\|^p\right] \\
&\quad + 3\mathbb{E}\left[\left\|\left(\nabla^2 f(\hat{x}_{t+1}, \hat{\xi}_{t+1}) - \nabla^2 F(\hat{x}_{t+1})\right) \cdot (x_{t+1} - x_t)\right\|^p\right] \\
&\leq 6L^p\gamma^p + 3\sigma_h^p\gamma^p.
\end{aligned}
$$

By the update rule for the gradient estimator:
$$\hat{e}_t \;=\; g_t - \nabla F(x_t) = (1-\alpha)\hat{e}_{t-1} + \alpha e_t + (1-\alpha)\hat{S}_t.$$

Unrolling the recursion, we have
$$\hat{e}_t \;=\; (1-\alpha)^t \hat{e}_0 + \alpha \sum_{j=0}^{t-1}(1-\alpha)^{t-j-1}e_{j+1} + \sum_{j=0}^{t-1}(1-\alpha)^{t-j}\hat{S}_{j+1}.$$

Then, taking the norm and the total expectation, we get
$$\mathbb{E}\left[\|\hat{e}_t\|\right] \le (1-\alpha)^t \mathbb{E}\left[\|\hat{e}_0\|\right] + \mathbb{E}\left[\left\|\alpha\sum_{j=0}^{t-1}(1-\alpha)^{t-j-1}e_{j+1}\right\|\right] + \mathbb{E}\left[\left\|\sum_{j=0}^{t-1}(1-\alpha)^{t-j}\hat{S}_{j+1}\right\|\right]. \quad (13)$$

To continue the proof, we need to bound the last two terms from the previous inequality. By Jensen's inequality, we obtain
$$\mathbb{E}\left[\left\|\alpha\sum_{j=0}^{t-1}(1-\alpha)^{t-j-1}e_{j+1}\right\|\right] \;\le\; \left(\mathbb{E}\left[\left\|\alpha\sum_{j=0}^{t-1}(1-\alpha)^{t-j-1}e_{j+1}\right\|^p\right]\right)^{1/p}$$

$$\overset{(*)}{\le} \left(2\alpha^p \sum_{j=0}^{t-1}(1-\alpha)^{p(t-j-1)}\mathbb{E}\left[\|e_{j+1}\|^p\right]\right)^{1/p}$$

$$\overset{(**)}{\le} \left(2\alpha^p \sum_{j=0}^{t-1}(1-\alpha)^{p(t-j-1)}\sigma^p\right)^{1/p}$$

$$\overset{(***)}{\le} 2\left(\alpha^{p-1}\sigma^p\right)^{1/p} = 2\alpha^{\frac{p-1}{p}}\sigma, \quad (14)$$

where in $(*)$ we used Lemma 1, in $(**)$ we used $\mathbb{E}\left[\|e_{j+1}\|^p\right] \le \sigma^p$ and in $(***)$ we used the following inequality
$$\sum_{j=0}^{t-1}(1-\alpha)^{p(t-j-1)} \le \sum_{j=0}^{t-1}(1-\alpha)^{t-j-1} \le \sum_{j=0}^{\infty}(1-\alpha)^j = \frac{1}{\alpha}.$$

We bound the third term in the same way as we did for the second term:
$$\mathbb{E}\left[\left\|\sum_{j=0}^{t-1}(1-\alpha)^{t-j}\hat{S}_{j+1}\right\|\right] \;\le\; \left(\mathbb{E}\left[\left\|\alpha\sum_{j=0}^{t-1}(1-\alpha)^{t-j}\hat{S}_{j+1}\right\|^p\right]\right)^{1/p}$$

$$\overset{(*)}{\le} \left(2\sum_{j=0}^{t-1}(1-\alpha)^{p(t-j)}\mathbb{E}\left[\left\|\hat{S}_{j+1}\right\|^p\right]\right)^{1/p}$$

$$\overset{(**)}{\le} \left(12\sum_{j=0}^{t-1}(1-\alpha)^{p(t-j)}\gamma^p(L^p + \sigma_h^p)\right)^{1/p}$$

$$\overset{(***)}{\le} 12\left(\alpha^{-1}\gamma^p(L^p + \sigma_h^p)\right)^{1/p} \le 12(L + \sigma_h)\gamma\alpha^{-1/p}, \quad (15)$$

where in $(*)$ we used Lemma 1, in $(**)$ we used $\mathbb{E}\left[\left\|\hat{S}_{j+1}\right\|^p\right] \le 6L^p\gamma^p + 3\sigma^p\gamma^p$, and in $(***)$ we used the following inequality
$$\sum_{j=0}^{t-1}(1-\alpha)^{p(t-j)} \le \sum_{j=0}^{t-1}(1-\alpha)^{t-j} \le \sum_{j=0}^{\infty}(1-\alpha)^j = \alpha^{-1}.$$

Plugging (14) and (15) into (13), we obtain
$$\mathbb{E}\left[\|\hat{e}_t\|\right] \le (1-\alpha)^t \mathbb{E}\left[\|\hat{e}_0\|\right] + 2\alpha^{\frac{p-1}{p}}\sigma + 12\gamma(L + \sigma_h)\alpha^{-1/p},$$

which concludes the proof. $\qquad\square$

**Main Theorem.** Now we are ready to state and to prove the main convergence theorem for Algorithm 1.

**Theorem 4.** *Let Assumptions 1, 2, 3 and 4 hold, and stepsize* $\gamma = \sqrt{\frac{\Delta\alpha^{1/p}}{(L+\sigma_h)T}}$, *momentum parameter* $\alpha = \min\{1, \alpha_{\text{eff}}\}$, *where* $\alpha_{\text{eff}} = \max\left\{\left(\frac{\mathcal{E}_0}{T\sigma}\right)^{\frac{p}{2p-1}}, \left(\frac{\Delta(L+\sigma_h)}{T\sigma^2}\right)^{\frac{p}{2p-1}}\right\}$. *Then iterates* $\{x_t\}_{t=0}^{T-1}$ *of Algorithm 1 satisfy*

$$\frac{1}{T}\sum_{t=0}^{T-1}\mathbb{E}\left[\|\nabla F(x_t)\|\right] = \mathcal{O}\left(\sqrt{\frac{\Delta(L+\sigma_h)}{T}} + \sigma\left(\frac{\Delta(L+\sigma_h)}{T\sigma^2}\right)^{\frac{p-1}{2p-1}} + \frac{\mathcal{E}_0}{T} + \sigma\left(\frac{\mathcal{E}_0}{T\sigma}\right)^{\frac{p-1}{2p-1}}\right),$$

*where* $\mathcal{E}_0$ *is defined as some upper bound on* $\mathbb{E}\left[\|g_0 - \nabla F(x_0)\|\right]$

*Proof.* Applying Lemma 9, we have

$$
\begin{aligned}
\sum_{t=0}^{T-1}\mathbb{E}\left[\|\hat{e}_t\|\right] &\leq \sum_{t=0}^{T-1}(1-\alpha)^t\mathbb{E}\left[\|\hat{e}_0\|\right] + 2\alpha^{\frac{p-1}{p}}\sigma T + 12\gamma(L+\sigma_h)\alpha^{-1/p}T \\
&\leq \frac{\mathbb{E}\left[\|\hat{e}_0\|\right]}{\alpha} + 2\alpha^{\frac{p-1}{p}}\sigma T + 12\gamma(L+\sigma_h)\alpha^{-1/p}T.
\end{aligned}
$$

Therefore, according to Lemma 8, we obtain

$$
\begin{aligned}
\frac{1}{T}\sum_{t=0}^{T-1}\mathbb{E}\left[\|\nabla F(x_t)\|\right] &\leq \frac{\Delta}{\gamma T} + \frac{2}{T}\sum_{t=0}^{T-1}\mathbb{E}\left[\|\hat{e}_t\|\right] + \frac{\gamma L}{2} \\
&\leq \frac{\Delta}{\gamma T} + \frac{\gamma L}{2} + \frac{2\mathbb{E}\left[\|\hat{e}_0\|\right]}{\alpha T} + 4\alpha^{\frac{p-1}{p}}\sigma + 24\gamma(L+\sigma_h)\alpha^{-1/p} \\
&\leq \frac{\Delta}{\gamma T} + \frac{2\mathbb{E}\left[\|\hat{e}_0\|\right]}{\alpha T} + 4\alpha^{\frac{p-1}{p}}\sigma + 25\gamma(L+\sigma_h)\alpha^{-1/p} \\
&\leq \mathcal{O}\left(\sqrt{\frac{\Delta(L+\sigma_h)}{\alpha^{1/p}T}} + \frac{\mathbb{E}\left[\|\hat{e}_0\|\right]}{\alpha T} + \alpha^{\frac{p-1}{p}}\sigma\right),
\end{aligned}
$$

where in the last inequality we took $\gamma = \sqrt{\frac{\Delta\alpha^{1/p}}{(L+\sigma_h)T}}$. Denoting $\mathbb{E}\left[\|\hat{e}_0\|\right] \leq \mathcal{E}_0$ and taking momentum parameter $\alpha = \min\{1, \alpha_{\text{eff}}\}$, where $\alpha_{\text{eff}} = \max\left\{\left(\frac{\mathcal{E}_0}{T\sigma}\right)^{\frac{p}{2p-1}}, \left(\frac{\Delta(L+\sigma_h)}{T\sigma^2}\right)^{\frac{p}{2p-1}}\right\}$, we have

$$
\begin{aligned}
\frac{1}{T}\sum_{t=0}^{T-1}\mathbb{E}\left[\|\nabla F(x_t)\|\right] &= \mathcal{O}\left(\sqrt{\frac{\Delta(L+\sigma_h)}{\alpha^{1/p}T}} + \frac{\mathcal{E}_0}{\alpha T} + \alpha^{\frac{p-1}{p}}\sigma\right) \\
&= \mathcal{O}\left(\sqrt{\frac{\Delta(L+\sigma_h)}{T}} + \frac{\mathcal{E}_0}{T} + \sigma\left(\frac{\Delta(L+\sigma_h)}{T\sigma^2}\right)^{\frac{p-1}{2p-1}} + \sigma\left(\frac{\mathcal{E}_0}{T\sigma}\right)^{\frac{p-1}{2p-1}}\right).
\end{aligned}
$$

$\square$

Now we investigate how different choices of initial estimator $g_0$ affect the total sample complexity bound.

**Corollary 2.** *Let all assumptions of Theorem 4 hold and the step-size and momentum parameters are set according to this theorem statement.*

1. *If we set* $g_0 = \nabla F(x_0)$, *then* $\mathcal{E}_0 = 0$, *and the total sample complexity of Algorithm 1 is equal to*

$$\mathcal{O}\left(\frac{\Delta(L+\sigma_h)}{\varepsilon^2} + \frac{\Delta(L+\sigma_h)}{\varepsilon^2}\left(\frac{\sigma}{\varepsilon}\right)^{\frac{1}{p-1}}\right).$$

2. *If we set $g_0 = 0$, then $\mathcal{E}_0 = \sqrt{2\Delta L}$, and the total sample complexity of Algorithm 1 is equal to*

$$\mathcal{O}\left(\frac{\Delta(L+\sigma_h)}{\varepsilon^2} + \frac{\Delta(L+\sigma_h)}{\varepsilon^2}\left(\frac{\sigma}{\varepsilon}\right)^{\frac{1}{p-1}} + \frac{\sqrt{\Delta L}\sigma}{\varepsilon^2}\left(\frac{\sigma}{\varepsilon}\right)^{\frac{1}{p-1}}\right).$$

3. *If we set $g_0 = \frac{1}{B_{init}}\sum_{j=1}^{B_{init}}\nabla f(x_0,\xi_{0,j})$ with $B_{init} = \max\left\{1, \left(\frac{\sigma}{\varepsilon}\right)^{\frac{p}{p-1}}, \left(\frac{\sigma}{\varepsilon}\right)^{\frac{p}{2p-1}}\right\}$, then $\mathcal{E}_0 = 2\sigma/B_{init}^{\frac{p-1}{p}}$, and the total sample complexity of Algorithm 1 is equal to*

$$\mathcal{O}\left(\frac{\Delta(L+\sigma_h)}{\varepsilon^2} + \frac{\Delta(L+\sigma_h)}{\varepsilon^2}\left(\frac{\sigma}{\varepsilon}\right)^{\frac{1}{p-1}} + \frac{\sigma}{\varepsilon}\left(\frac{\sigma}{\varepsilon}\right)^{\frac{1}{p-1}} + \left(\frac{\sigma}{\varepsilon}\right)^{\frac{p}{2p-1}}\right).$$

*Proof.* The first case is ideal, when we can have access to the full gradient but once: set $g_0 = \nabla F(x_0)$, then $\hat{e}_0 = 0$ and $\mathcal{E}_0 = 0$. Thus we have

$$\frac{1}{T}\sum_{t=0}^{T-1}\mathbb{E}\left[\|\nabla F(x_t)\|\right] = \mathcal{O}\left(\sqrt{\frac{\Delta(L+\sigma_h)}{T}} + \sigma\left(\frac{\Delta(L+\sigma_h)}{T\sigma^2}\right)^{\frac{p-1}{2p-1}}\right).$$

In other words, we can guarantee $\frac{1}{T}\sum_{t=0}^{T-1}\mathbb{E}\left[\|\nabla F(x_t)\|\right] \le \varepsilon$ after

$$\mathcal{O}\left(\frac{\Delta(L+\sigma_h)}{\varepsilon^2} + \frac{\Delta(L+\sigma_h)}{\varepsilon^2}\left(\frac{\sigma}{\varepsilon}\right)^{\frac{1}{p-1}}\right).$$

iterations of Algorithm 1.

The second case is that we select $g_0$ as a zero vector. This choice implies that by smoothness of $F$, we have

$$\mathbb{E}\left[\|\hat{e}_0\|\right] = \|\nabla F(x_0)\| \le \sqrt{2L\Delta} = \mathcal{E}_0.$$

Plugging the obtained value of $\mathcal{E}_0$ into the result of Theorem 2, we have

$$\frac{1}{T}\sum_{t=0}^{T-1}\mathbb{E}\left[\|\nabla F(x_t)\|\right] = \mathcal{O}\left(\sqrt{\frac{\Delta(L+\sigma_h)}{T}} + \sigma\left(\frac{\Delta(L+\sigma_h)}{T\sigma^2}\right)^{\frac{p-1}{2p-1}} + \frac{\sqrt{\Delta L}}{T} + \sigma\left(\frac{\sqrt{\Delta L}}{T\sigma}\right)^{\frac{p-1}{2p-1}}\right)$$

$$= \mathcal{O}\left(\sqrt{\frac{\Delta(L+\sigma_h)}{T}} + \sigma\left(\frac{\Delta(L+\sigma_h)}{T\sigma^2}\right)^{\frac{p-1}{2p-1}} + \sigma\left(\frac{\sqrt{\Delta L}}{T\sigma}\right)^{\frac{p-1}{2p-1}}\right)$$

which implies that the total sample complexity is

$$\mathcal{O}\left(\frac{\Delta(L+\sigma_h)}{\varepsilon^2} + \frac{\Delta(L+\sigma_h)}{\varepsilon^2}\left(\frac{\sigma}{\varepsilon}\right)^{\frac{1}{p-1}} + \frac{\sqrt{\Delta L}\sigma}{\varepsilon^2}\left(\frac{\sigma}{\varepsilon}\right)^{\frac{1}{p-1}}\right).$$

The third case is that we set $g_0 = \frac{1}{B_{init}}\sum_{j=1}^{B_{init}}\nabla f(x_0,\xi_{0,j})$. By Lemma 1, we have

$$
\begin{aligned}
\mathbb{E}\left[\|\hat{e}_0\|\right] &\le \left(\mathbb{E}\left[\|\hat{e}_0\|^p\right]\right)^{1/p} = \left(\mathbb{E}\left[\|g_0 - \nabla F(x_0)\|^p\right]\right)^{1/p} \\
&\le \frac{2}{B_{init}}\left(\sum_{j=0}^{B_{init}}\mathbb{E}\left[\|\nabla f(x_0,\xi_{0,j}) - \nabla F(x_0)\|^p\right]\right)^{1/p} \\
&\le \frac{2}{B_{init}}\left(\sum_{j=0}^{B_{init}}\sigma^p\right)^{1/p} = \frac{2\sigma}{B_{init}^{\frac{p-1}{p}}} = \mathcal{E}_0.
\end{aligned}
$$

Plugging the obtained value of $\mathcal{E}_0$ into the result of Theorem 2, we have

$$\frac{1}{T}\sum_{t=0}^{T-1}\mathbb{E}\left[\|\nabla F(x_t)\|\right] = \mathcal{O}\left(\sqrt{\frac{\Delta(L+\sigma_h)}{T}} + \sigma\left(\frac{\Delta(L+\sigma_h)}{T\sigma^2}\right)^{\frac{p-1}{2p-1}} + \frac{\sigma}{TB_{init}^{\frac{p-1}{p}}} + \sigma\left(\frac{1}{TB_{init}^{\frac{p-1}{p}}}\right)^{\frac{p-1}{2p-1}}\right),$$

from which we have that the total sample complexity is

$$\mathcal{O}\left(\frac{\Delta(L+\sigma_h)}{\varepsilon^2} + \frac{\Delta(L+\sigma_h)}{\varepsilon^2}\left(\frac{\sigma}{\varepsilon}\right)^{\frac{1}{p-1}} + \frac{\sigma^2}{B_{\text{init}}^{\frac{p-1}{p}}\varepsilon^2}\left(\frac{\sigma}{\varepsilon}\right)^{\frac{1}{p-1}} + \frac{\sigma}{\varepsilon B_{\text{init}}^{\frac{p-1}{p}}} + B_{\text{init}}\right).$$

Taking $B_{\text{init}} = \max\left\{1, \left(\frac{\sigma}{\varepsilon}\right)^{\frac{p}{p-1}}, \left(\frac{\sigma}{\varepsilon}\right)^{\frac{p}{2p-1}}\right\}$, we have that the total sample complexity is

$$\mathcal{O}\left(\frac{\Delta(L+\sigma_h)}{\varepsilon^2} + \frac{\Delta(L+\sigma_h)}{\varepsilon^2}\left(\frac{\sigma}{\varepsilon}\right)^{\frac{1}{p-1}} + \frac{\sigma}{\varepsilon}\left(\frac{\sigma}{\varepsilon}\right)^{\frac{1}{p-1}} + \left(\frac{\sigma}{\varepsilon}\right)^{\frac{p}{2p-1}}\right).$$

$\square$

# F  Missing Proofs for Section 5

Now we start the high-probability convergence analysis for Algorithm 2. We divide our analysis into two parts: Optimization part, where we prove descent lemma, and High-Probability part, where we use concentration inequality to bound several terms from Descent Lemma. Finally, combining results from both parts, we prove the main results of this section via induction. The idea is based on work of Sadiev et al. [2023] and Liu et al. [2023b].

## F.1  Analysis: Optimization Part

We start with *Descent Lemma*.

**Lemma 10.** *Let Assumptions 1, 2 hold. Then Algorithm 2 with stepsize $\gamma > 0$ and momentum parameter $\alpha \in (0, 1)$ generates iterates $\{x_t\}_{t=0}^T$ satisfying the following inequality*

$$\gamma \sum_{t=0}^{T-1} \|\nabla F(x_t)\| + \Delta_T \leq \Delta_1 + \frac{\gamma^2 LT}{2} + \frac{3\gamma}{\alpha}\sqrt{L\Delta_1}$$

$$+ 2\gamma\alpha \sum_{t=1}^{T-1} \left\|\sum_{j=1}^t (1-\alpha)^{t-j}\theta_j\right\| + 2\gamma \sum_{t=1}^{T-1}\left\|\sum_{j=1}^t (1-\alpha)^{t-j+1}\omega_j\right\|,$$

*for any $j \in [T-1]$ vectors $\theta_j$ and $\omega_j$ are defined as follows*

$$\theta_j \stackrel{def}{=} \texttt{clip}\left(\nabla f(x_j, \xi_j), \lambda\right) - \nabla F(x_j), \tag{16}$$

$$\omega_j \stackrel{def}{=} \texttt{clip}\left(\nabla^2 f(\hat{x}_t, \hat{\xi}_t)(x_t - x_{t-1}), \lambda_h\right) - \left(\nabla F(x_j) - \nabla F(x_{j-1})\right). \tag{17}$$

*Proof.* First, we notice that Lemma 8 holds for Algorithm 2 and can be proven in the same way as for Algorithm 1, since the gradient updates in the both methods are identical except for $x_1$, and the update rule for momentum term does not play any role in the proof of Lemma 8. Therefore, the iterates $\{x_t\}_{t=0}^T$ of Algorithm 2 satisfy

$$\gamma \sum_{t=0}^{T-1}\|\nabla F(x_t)\| + \Delta_T \leq \Delta_1 + 2\gamma\sum_{t=0}^{T-1}\|\hat{e}_t\| + \frac{\gamma^2 LT}{2}, \tag{18}$$

where $\Delta_0 = \Delta_1$, because $x_0 = x_1$. Next we bound $\|\hat{e}_t\|$ in almost the same way as we did in Lemma 9. By update rule for momentum parameter, we have

$$\begin{aligned}
\hat{e}_t &= g_t - \nabla F(x_t) \\
&= (1-\alpha)\left(g_{t-1} + \texttt{clip}\left(\nabla^2 f(\hat{x}_t, \hat{\xi}_t)(x_t - x_{t-1}), \lambda_h\right)\right) \\
&\quad + \alpha\texttt{clip}\left(\nabla f(x_t, \xi_t), \lambda\right) - \nabla F(x_t) \\
&\stackrel{(16),(17)}{=} (1-\alpha)\hat{e}_{t-1} + \alpha\theta_t + (1-\alpha)\omega_t \\
&= (1-\alpha)^t\hat{e}_0 + \alpha\sum_{j=1}^t (1-\alpha)^{t-j}\theta_j + \sum_{j=1}^t (1-\alpha)^{t-j+1}\omega_j.
\end{aligned}$$

Taking the norm, we obtain the following bound

$$\begin{aligned}
\|\hat{e}_t\| &\leq (1-\alpha)^t\|\hat{e}_0\| + \alpha\left\|\sum_{j=1}^t (1-\alpha)^{t-j}\theta_j\right\| + \left\|\sum_{j=1}^t (1-\alpha)^{t-j+1}\omega_j\right\| \\
&\leq (1-\alpha)^t\sqrt{2L\Delta_1} + \alpha\left\|\sum_{j=1}^t (1-\alpha)^{t-j}\theta_j\right\| + \left\|\sum_{j=1}^t (1-\alpha)^{t-j+1}\omega_j\right\|, \tag{19}
\end{aligned}$$

where in the last inequality we used the following chain of inequalities

$$\|\hat{e}_0\| = \|g_0 - \nabla F(x_0)\| = \|\nabla F(x_0)\| = \|\nabla F(x_1)\| \leq \sqrt{2L(F(x_1) - F_*)} \leq \sqrt{2L\Delta_1}.$$

Plugging (19) into (18), we acquire

$$
\begin{aligned}
\gamma \sum_{t=0}^{T-1} \|\nabla F(x_t)\| + \Delta_T \quad \le \quad & \Delta_1 + 2\gamma \sum_{t=0}^{T-1} \|\hat{e}_t\| + \frac{\gamma^2 LT}{2} \\
\le \quad & \Delta_1 + \frac{\gamma^2 LT}{2} + 2\gamma \sum_{t=0}^{T-1} (1-\alpha)^t \sqrt{2L\Delta_1} \\
& + 2\gamma\alpha \sum_{t=1}^{T-1} \left\| \sum_{j=1}^{t} (1-\alpha)^{t-j} \theta_j \right\| + 2\gamma \sum_{t=1}^{T-1} \left\| \sum_{j=1}^{t} (1-\alpha)^{t-j+1} \omega_j \right\| \\
\le \quad & \Delta_1 + \frac{\gamma^2 LT}{2} + \frac{3\gamma}{\alpha} \sqrt{L\Delta_1} \\
& + 2\gamma\alpha \sum_{t=1}^{T-1} \left\| \sum_{j=1}^{t} (1-\alpha)^{t-j} \theta_j \right\| + 2\gamma \sum_{t=1}^{T-1} \left\| \sum_{j=1}^{t} (1-\alpha)^{t-j+1} \omega_j \right\|,
\end{aligned}
$$

where in the last inequality we used

$$
2\gamma \sum_{t=0}^{T-1} (1-\alpha)^t \sqrt{2L\Delta_1} \le 2\gamma \sum_{t=0}^{\infty} (1-\alpha)^t \sqrt{2L\Delta_1} \le \frac{2\sqrt{2}\gamma}{1-(1-\alpha)} \sqrt{L\Delta_1} \le \frac{3\gamma}{\alpha} \sqrt{L\Delta_1}.
$$

$\square$

## F.2 Analysis: Statistical Part

According to Lemma 10, we have two new terms

$$
\left\| \sum_{j=1}^{t} (1-\alpha)^{t-j} \theta_j \right\| \quad \text{and} \quad \left\| \sum_{j=1}^{t} (1-\alpha)^{t-j+1} \omega_j \right\|.
$$

To bound both of them, we use the same idea as in the work of Gorbunov et al. [2020], Sadiev et al. [2023], Liu et al. [2023b]: introduce unbiased and biased parts of $\theta_t$ and $\omega_t$, i.e. for any $t \in [T]$

$$
\begin{aligned}
\theta_t \;&=\; \theta_t^u + \theta_t^b, \quad \text{where} \\
\theta_t^u \;&\stackrel{\text{def}}{=}\; \texttt{clip}\left(\nabla f(x_t, \xi_t), \lambda\right) - \mathbb{E}_{\xi_t}\left[\texttt{clip}\left(\nabla f(x_t, \xi_t), \lambda\right)\right], & (20) \\
\theta_t^b \;&\stackrel{\text{def}}{=}\; \mathbb{E}_{\xi_t}\left[\texttt{clip}\left(\nabla f(x_t, \xi_t), \lambda\right)\right] - \nabla F(x_t); & (21) \\
\omega_t \;&=\; \omega_t^u + \omega_t^b, \quad \text{where} \\
\omega_t^u \;&\stackrel{\text{def}}{=}\; \texttt{clip}\left(\nabla^2 f(\hat{x}_t, \hat{\xi}_t)(x_t - x_{t-1}), \lambda_h\right) - \mathbb{E}_{q_t, \xi_t}\left[\texttt{clip}\left(\nabla^2 f(\hat{x}_t, \hat{\xi}_t)(x_t - x_{t-1}), \lambda_h\right)\right] & (22) \\
\omega_t^b \;&\stackrel{\text{def}}{=}\; \mathbb{E}_{q_t, \xi_t}\left[\texttt{clip}\left(\nabla^2 f(\hat{x}_t, \hat{\xi}_t)(x_t - x_{t-1}), \lambda_h\right)\right] - \left(\nabla F(x_j) - \nabla F(x_{j-1})\right). & (23)
\end{aligned}
$$

Under Assumption 3 for any $\lambda \ge 2\|\nabla F(x_t)\|$ Lemma 4 implies

$$
\mathbb{E}_{\xi_t}\left[\|\theta_j^u\|^2\right] \le 18\lambda^{2-p}\sigma^p \quad \text{and} \quad \|\theta_t^b\| \le 2^p \lambda^{1-p}\sigma^p, \quad \text{for all } t \in [T]. \tag{24}
$$

It is worth to mention we have already shown in Lemma 8 that under Assumptions 2 and 4 the vector $\hat{S}_t = \nabla^2 f(\hat{x}_t, \hat{\xi}_t) \cdot (x_t - x_{t-1}) - \nabla F(x_t) + \nabla F(x_{t-1})$ has zero expectation $\mathbb{E}_{q_t, \xi_t}\left[\hat{S}_t\right] = 0$, since

$$
\begin{aligned}
\mathbb{E}_{q_t, \hat{\xi}_t}\left[\nabla^2 f(\hat{x}_t, \hat{\xi}_t)(x_t - x_{t-1})\right] \;&=\; \mathbb{E}_{q_t}\left[\mathbb{E}_{\hat{\xi}_t}\left[\nabla^2 f(\hat{x}_t, \hat{\xi}_t)\right](x_t - x_{t-1})\right] \\
&=\; \mathbb{E}_{q_t}\left[\nabla^2 F(\hat{x}_t)(x_t - x_{t-1})\right] \\
&=\; \int_0^1 \nabla^2 F(qx_t + (1-q)x_{t-1})(x_t - x_{t-1})dq \\
&=\; \nabla F(x_t) - \nabla F(x_{t-1}).
\end{aligned}
$$

The $p$-central moment of $\hat{S}_t$ is bounded, i.e. $\mathbb{E}_{q_t,\xi_t}\left[\left\|\hat{S}_t\right\|^p\right] \leq 6L^p\gamma^p + 3\sigma_h^p\gamma^p$. Then, according to Lemma 4, for all $t \in [T]$ we have

$$\mathbb{E}_{q_t,\xi_t}\left[\left\|\omega_j^u\right\|^2\right] \leq 18\lambda_h^{2-p}\gamma^p(6L^p + 3\sigma_h^p) \quad \text{and} \quad \left\|\omega_t^b\right\| \leq 2^p\lambda_h^{1-p}\gamma^p(6L^p + 3\sigma_h^p), \qquad (25)$$

for any $2\left\|\nabla F(x_t) - \nabla F(x_{t-1})\right\| \leq 2L\gamma \leq \lambda_h$.

**Lemma 11.** *Let Assumption 3 hold. For any $\delta' \in (0, 1/2]$ and any $t \in [T]$, if clipping level satisfy*

$$\lambda \geq \max\left\{2\|\nabla F(x_j)\|, \frac{\sigma}{\alpha^{1/p}}\right\},$$

*then with probability at least $1 - 2\delta'$*

$$\left\|\sum_{j=1}^{t}(1-\alpha)^{t-j}\theta_j\right\| \leq 22\lambda\log\frac{2}{\delta'}.$$

*Proof.* We start with bounding $\left\|\sum_{j=1}^{t}(1-\alpha)^{t-j}\theta_j\right\|$.

**Upper bound for** $\left\|\sum_{j=1}^{t}(1-\alpha)^{t-j}\theta_j\right\|$    By (20) and (21), we have

$$\left\|\sum_{j=1}^{t}(1-\alpha)^{t-j}\theta_j\right\| \leq \left\|\sum_{j=1}^{t}(1-\alpha)^{t-j}\theta_j^u\right\| + \underbrace{\left\|\sum_{j=1}^{t}(1-\alpha)^{t-j}\theta_j^b\right\|}_{④}.$$

Denote $Y_j^t \overset{\text{def}}{=} \left\|(1-\alpha)^{t-j}\theta_j^u\right\|^2 - \mathbb{E}_{\xi_j}\left[\left\|(1-\alpha)^{t-j}\theta_j^u\right\|^2\right]$, and $|X_j^t| \leq \left\|(1-\alpha)^{t-j}\theta_j^u\right\|$ for any $j \in [t]$

$$V_j^t \overset{\text{def}}{=} \begin{cases} 0, & \text{if } j = 0; \\ \text{sign}\left(\sum_{i=1}^{j-1}V_i^t\right)\frac{\left\langle\sum_{i=1}^{j-1}(1-\alpha)^{t-i}\theta_i^u,(1-\alpha)^{t-j}\theta_j^u\right\rangle}{\left\|\sum_{i=1}^{j-1}(1-\alpha)^{t-i}\theta_i^u\right\|}, & \text{if } j \neq 0 \text{ and } \sum_{i=1}^{j-1}(1-\alpha)^{t-i}\theta_i^u \neq 0; \\ 0, & \text{if } j \neq 0 \text{ and } \sum_{i=1}^{j-1}(1-\alpha)^{t-i}\theta_i^u = 0. \end{cases}$$

Then to bound $\left\|\sum_{j=1}^{t}(1-\alpha)^{t-j}\theta_j^u\right\|$ we use Lemma 2 and obtain

$$\begin{aligned}\left\|\sum_{j=1}^{t}(1-\alpha)^{t-j}\theta_j^u\right\| &\leq \left|\sum_{j=1}^{t}V_j^t\right| + \sqrt{\max_{j\in[t]}\left\|(1-\alpha)^{t-j}\theta_j^u\right\|^2 + \sum_{j=1}^{t}\left\|(1-\alpha)^{t-j}\theta_j^u\right\|^2} \\ &\leq \left|\sum_{j=1}^{t}V_j^t\right| + \sqrt{2\sum_{j=1}^{t}\left\|(1-\alpha)^{t-j}\theta_j^u\right\|^2} \\ &\leq \underbrace{\left|\sum_{j=1}^{t}V_j^t\right|}_{①} + \left(2\cdot\underbrace{\sum_{j=1}^{t}Y_j^t}_{②} + 2\cdot\underbrace{\sum_{j=1}^{t}\mathbb{E}_{\xi_j}\left[\left\|(1-\alpha)^{t-j}\theta_j^u\right\|^2\right]}_{③}\right)^{1/2}.\end{aligned}$$

**Upper bound for** ①**.** The sequence $X_1^t, \ldots, X_t^t$ is martingale difference sequence, since, by definition of $\theta_j^t$ and $X_j^t$, for all $j \in [t]$ we have $\mathbb{E}\left[X_j^t \mid X_{j-1}^t, \ldots, X_1^t\right] = \mathbb{E}_{\xi_j}\left[X_j^t\right] = 0$. Also, according to Lemma 2, we have $|X_j^t| \leq \left\|(1-\alpha)^{t-j}\theta_j^u\right\|$ for all $j \in [t]$. Using Lemma 4, we obtain that

$$|X_j^t| \leq \left\|(1-\alpha)^{t-j}\theta_j^u\right\| \leq \left\|\theta_j^u\right\| \leq 2\lambda, \quad \text{where } c_1 \overset{\text{def}}{=} 2\lambda. \qquad (26)$$

Denoting $\sigma_j^2 \overset{\text{def}}{=} \mathbb{E}\left[(X_j^t)^2 \mid X_{j-1}^t, \ldots, X_1^t\right] = \mathbb{E}_{\xi_j}\left[(X_j^t)^2\right]$, Lemma 3 implies

$$\mathbb{P}\left\{|①| > b_1 \text{ and } \sum_{j=1}^{t} \sigma_j^2 \le G_1 \log \frac{2}{\delta'}\right\} \le 2\exp\left(-\frac{b_1^2}{2G_1 \log \frac{2}{\delta'} + \frac{2c_1 b_1}{3}}\right) = \delta',$$

where the last identity is true, if we set $b_1 = \left(\frac{1}{3}c_1 + \sqrt{\frac{1}{9}c_1 + 2G_1}\right)\log \frac{2}{\delta'}$. To define $G_1$, we need to bound $\sum_{j=1}^{t} \sigma_j^2$:

$$
\begin{aligned}
\sum_{j=1}^{t} \sigma_j^2 &= \sum_{j=1}^{t} \mathbb{E}_{\xi_j}\left[(X_j^t)^2\right] \le \sum_{j=1}^{t} \mathbb{E}_{\xi_j}\left[\|(1-\alpha)^{t-j}\theta_j^u\|^2\right] = \sum_{j=1}^{t}(1-\alpha)^{2(t-j)}\mathbb{E}_{\xi_j}\left[\|\theta_j^u\|^2\right] \\
&\overset{(24)}{\le} \sum_{j=1}^{t}(1-\alpha)^{2(t-j)} \cdot 18\lambda^{2-p}\sigma^p \le \frac{18\lambda^{2-p}\sigma^p}{1-(1-\alpha)^2} \le \frac{18\lambda^{2-p}\sigma^p}{\alpha} = G_1.
\end{aligned}
\tag{27}
$$

where we assumed $\|\nabla F(x_j)\| \le \frac{\lambda}{2}$ for any $j \in [t]$.

**Upper bound for ②.** The sequence $Y_1^t, \ldots, Y_t^t$ forms a martingale difference sequence, since the definition of $Y_j^t$ leads to $\mathbb{E}\left[Y_j^t \mid Y_{j-1}^t, \ldots, Y_1^t\right] = \mathbb{E}_{\xi_j}\left[Y_j^t\right] = 0$ for all $j \in [t]$. Also, according to Lemma 4, we obtain that

$$|Y_j^t| \le \left\|(1-\alpha)^{t-j}\theta_j^u\right\|^2 + \mathbb{E}_{\xi_j}\left[\left\|(1-\alpha)^{t-j}\theta_j^u\right\|^2\right] \le 4\lambda^2 + 4\lambda^2 = 8\lambda^2, \tag{28}$$

where we define $c_2 \overset{\text{def}}{=} 8\lambda^2$. Denoting the conditional variance of $Y_j^t$ as $\widetilde{\sigma}_j^2 \overset{\text{def}}{=} \mathbb{E}\left[(Y_j^t)^2 \mid Y_{j-1}^t, \ldots, Y_1^t\right] = \mathbb{E}_{\xi_j}\left[(Y_j^t)^2\right]$, we can bound $\widetilde{\sigma}_j^2$ easily

$$
\begin{aligned}
\widetilde{\sigma}_j^2 &\overset{(28)}{\le} 8\lambda^2 \cdot \mathbb{E}\left[\left|\left\|(1-\alpha)^{t-j}\theta_j^u\right\|^2 - \mathbb{E}_{\xi_j}\left[\left\|(1-\alpha)^{t-j}\theta_j^u\right\|^2\right]\right|\right] \\
&\le 16\lambda^2 \mathbb{E}_{\xi_j}\left[\left\|(1-\alpha)^{t-j}\theta_j^u\right\|^2\right].
\end{aligned}
$$

Then, Lemma 3 implies

$$\mathbb{P}\left\{|②| > b_2 \text{ and } \sum_{j=1}^{t} \widetilde{\sigma}_j^2 \le G_2 \log \frac{2}{\delta'}\right\} \le 2\exp\left(-\frac{b_2^2}{2G_2 \log \frac{2}{\delta'} + \frac{2c_2 b_2}{3}}\right) = \delta',$$

where the last identity is true, if we set $b_2 = \left(\frac{1}{3}c_2 + \sqrt{\frac{1}{9}c_2 + 2G_2}\right)\log \frac{2}{\delta'}$. To define $G_2$, we need to bound $\sum_{j=1}^{t} \widetilde{\sigma}_j^2$:

$$
\begin{aligned}
\sum_{j=1}^{t} \widetilde{\sigma}_j^2 &\le 16\lambda^2 \sum_{j=1}^{t} \mathbb{E}_{\xi_j}\left[\left\|(1-\alpha)^{t-j}\theta_j^u\right\|^2\right] \\
&= 16\lambda^2 \sum_{j=1}^{t}(1-\alpha)^{2(t-j)}\mathbb{E}_{\xi_j}\left[\left\|\theta_j^u\right\|^2\right] \\
&\overset{(27)}{\le} 16\lambda^2 \cdot \frac{18\lambda^{2-p}\sigma^p}{\alpha} = \frac{16 \cdot 18\lambda^{4-p}\sigma^p}{\alpha} = G_2,
\end{aligned}
$$

where we assumed $\|\nabla F(x_j)\| \le \frac{\lambda}{2}$ for any $j \in [t]$.

**Upper bound for ③.** Thanks to the proof of the bound on ③ (see (27)), we have already shown what we need: with probability 1

$$③ = \sum_{j=1}^{t} \mathbb{E}_{\xi_j}\left[\left\|(1-\alpha)^{t-j}\theta_j^u\right\|^2\right] \overset{(27)}{\le} \frac{18\lambda^{2-p}\sigma^p}{\alpha}.$$

**Upper bound on ④.** Assuming $\|\nabla F(x_j)\| \le \frac{\lambda}{2}$ for any $j \in [t]$, with probability 1 we have

$$④ = \left\| \sum_{j=1}^{t} (1-\alpha)^{t-j} \theta_j^b \right\| \le \sum_{j=1}^{t} (1-\alpha)^{t-j} \|\theta_j^b\| \overset{(27)}{\le} 4\lambda^{1-p}\sigma^p \sum_{j=1}^{t} (1-\alpha)^{t-j} \le \frac{4\lambda^{1-p}\sigma^p}{\alpha}.$$

To sum up, we introduce event $E_{①,t}$ as follows

$$E_{①,t} \overset{\text{def}}{=} \left\{ |①| \le b_1 \text{ or } \sum_{j=1}^{t} \sigma_j^2 > G_1 \log \frac{2}{\delta'} \right\}, \tag{29}$$

where $c_1 = 2\lambda$, $G_1 = \frac{18\lambda^{2-p}\sigma^p}{\alpha}$, $b_1 = \left(\frac{1}{3}c_1 + \sqrt{\frac{1}{9}c_1^2 + 2G_1}\right) \log \frac{2}{\delta'}$. We have shown that $\mathbb{P}\{E_{①,t}\} \ge 1 - \delta'$. The bound on $b_1$

$$
\begin{aligned}
b_1 &= \left(\frac{1}{3}c_1 + \sqrt{\frac{1}{9}c_1^2 + 2G_1}\right) \log \frac{2}{\delta'} \le \left(\frac{2}{3}c_1 + \sqrt{2G_1}\right) \log \frac{2}{\delta'} \\
&= \left(\frac{4}{3}\lambda + \sqrt{36\frac{\lambda^{2-p}\sigma^p}{\alpha}}\right) \log \frac{2}{\delta'} = \lambda\left(\frac{4}{3} + 6\sqrt{\frac{1}{\alpha}\left(\frac{\sigma}{\lambda}\right)^p}\right) \log \frac{2}{\delta'}.
\end{aligned}
$$

Also we define event $E_{②,t}$ as follows

$$E_{②,t} = \left\{ |②| \le b_2 \text{ or } \sum_{j=1}^{t} \tilde{\sigma}_j^2 > G_2 \log \frac{2}{\delta'} \right\}, \tag{30}$$

where $c_2 = 8\lambda^2$, $G_2 = \frac{16 \cdot 18\lambda^{4-p}\sigma^p}{\alpha}$, $b_2 = \left(\frac{1}{3}c_2 + \sqrt{\frac{1}{9}c_2^2 + 2G_2}\right) \log \frac{2}{\delta'}$. We have shown that $\mathbb{P}\{E_{②,t}\} \ge 1 - \delta'$. We adjust the bound on $b_2$:

$$
\begin{aligned}
b_2 &= \left(\frac{1}{3}c_2 + \sqrt{\frac{1}{9}c_2^2 + 2G_2}\right) \log \frac{2}{\delta'} \le \left(\frac{2}{3}c_2 + \sqrt{2G_2}\right) \log \frac{2}{\delta'} \\
&= \left(\frac{16}{3}\lambda^2 + \sqrt{\frac{16 \cdot 36\lambda^{4-p}\sigma^p}{\alpha}}\right) \log \frac{2}{\delta'} = \lambda^2\left(\frac{16}{3} + 24\sqrt{\frac{1}{\alpha}\left(\frac{\sigma}{\lambda}\right)^p}\right) \log \frac{2}{\delta'}.
\end{aligned}
$$

Thus, the event $E_{①,t} \cap E_{②,t}$ implies

$$
\begin{aligned}
\left\| \sum_{j=1}^{t} (1-\alpha)^{t-j} \theta_j \right\| &\le \left\| \sum_{j=1}^{t} (1-\alpha)^{t-j} \theta_j^u \right\| + \left\| \sum_{j=1}^{t} (1-\alpha)^{t-j} \theta_j^b \right\| \\
&\le |①| + \sqrt{2 \cdot ② + 2 \cdot ③} + ④ \\
&\le \lambda\left(\frac{4}{3} + 6\sqrt{\frac{1}{\alpha}\left(\frac{\sigma}{\lambda}\right)^p}\right) \log \frac{2}{\delta'} + 4\lambda \cdot \frac{1}{\alpha}\left(\frac{\sigma}{\lambda}\right)^p \\
&\quad + \sqrt{2\lambda^2\left(\frac{16}{3} + 24\sqrt{\frac{1}{\alpha}\left(\frac{\sigma}{\lambda}\right)^p}\right) \log \frac{2}{\delta'} + \frac{36\lambda^{2-p}\sigma^p}{\alpha}}.
\end{aligned}
$$

Taking $\lambda \ge \frac{\sigma}{\alpha^{1/p}}$, we have the event $E_{①,t} \cap E_{②,t}$ implies

$$\left\| \sum_{j=1}^{t} (1-\alpha)^{t-j} \theta_j \right\| \le \lambda\left(\frac{4}{3} + 6 + 4 + \sqrt{\frac{32}{3} + 84}\right) \log \frac{2}{\delta'} \le 22\lambda \log \frac{2}{\delta'}.$$

This concludes the proof. $\qquad\square$

**Lemma 12.** *Let Assumptions 2and 4 hold. For any $\delta'' \in (0, 1/2]$ and any $t \in [T]$, if clipping level satisfy*

$$\lambda_h \geq \max\left\{2\gamma L, \frac{\gamma(L + \sigma_h)}{\alpha^{1/p}}\right\},$$

*and with probability at least $1 - 2\delta''$*

$$\left\|\sum_{j=1}^t (1-\alpha)^{t-j+1}\omega_j\right\| \leq 59\lambda_h \log\frac{2}{\delta''}.$$

*Proof.* By (22) and (23), we have

$$\left\|\sum_{j=1}^t (1-\alpha)^{t-j+1}\omega_j\right\| \leq \left\|\sum_{j=1}^t (1-\alpha)^{t-j+1}\omega_j^u\right\| + \underbrace{\left\|\sum_{j=1}^t (1-\alpha)^{t-j+1}\omega_j^b\right\|}_{\text{⑧}}. \qquad (31)$$

Denote $Z_j^t \overset{\text{def}}{=} \left\|(1-\alpha)^{t-j+1}\omega_j^u\right\|^2 - \mathbb{E}_{q_j,\xi_j}\left[\left\|(1-\alpha)^{t-j+1}\omega_j^u\right\|^2\right]$, and $|W_j^t| \leq \left\|(1-\alpha)^{t-j+1}\omega_j^u\right\|$ for any $j \in [t]$

$$W_j^t \overset{\text{def}}{=} \begin{cases} 0, & \text{if } j = 0; \\ \text{sign}\left(\sum\limits_{i=1}^{j-1} W_i^t\right) \dfrac{\left\langle \sum\limits_{i=1}^{j-1}(1-\alpha)^{t-i+1}\omega_i^u, (1-\alpha)^{t-j+1}\omega_j^u\right\rangle}{\left\|\sum\limits_{i=1}^{j-1}(1-\alpha)^{t-i+1}\omega_i^u\right\|}, & \text{if } j \neq 0 \text{ and } \sum\limits_{i=1}^{j-1}(1-\alpha)^{t-i+1}\omega_i^u \neq 0; \\ 0, & \text{if } j \neq 0 \text{ and } \sum\limits_{i=1}^{j-1}(1-\alpha)^{t-i+1}\omega_i^u = 0. \end{cases}$$

Then to bound $\left\|\sum\limits_{j=1}^t (1-\alpha)^{t-j+1}\omega_j^u\right\|$ we use Lemma 2 and obtain

$$\left\|\sum_{j=1}^t (1-\alpha)^{t-j}\omega_j^u\right\| \leq \left|\sum_{j=1}^t W_j^t\right| + \sqrt{\max_{j\in[t]}\left\|(1-\alpha)^{t-j}\omega_j^u\right\|^2 + \sum_{j=1}^t\left\|(1-\alpha)^{t-j}\omega_j^u\right\|^2}$$

$$\leq \left|\sum_{j=1}^t W_j^t\right| + \sqrt{2\sum_{j=1}^t\left\|(1-\alpha)^{t-j+1}\omega_j^u\right\|^2}$$

$$\leq \underbrace{\left|\sum_{j=1}^t W_j^t\right|}_{\text{⑤}} + \left(2\cdot\underbrace{\sum_{j=1}^t Z_j^t}_{\text{⑥}} + 2\cdot\underbrace{\sum_{j=1}^t\mathbb{E}_{q_j,\xi_j}\left[\left\|(1-\alpha)^{t-j+1}\omega_j^u\right\|^2\right]}_{\text{⑦}}\right)^{1/2}.$$

**Upper bound for ⑤.** The sequence $W_1^t, \ldots, W_t^t$ is martingale difference sequence, since, by definition of $\omega_j^t$ and $W_j^t$, for all $j \in [t]$ we have $\mathbb{E}\left[W_j^t \mid W_{j-1}^t, \ldots, W_1^t\right] = \mathbb{E}_{q_j,\xi_j}\left[W_j^t\right] = 0$. Also, according to Lemma 2, we have $|W_j^t| \leq \left\|(1-\alpha)^{t-j}\omega_j^u\right\|$ for all $j \in [t]$. Using Lemma 4, we obtain that

$$|W_j^t| \leq \left\|(1-\alpha)^{t-j}\omega_j^u\right\| \leq \left\|\omega_j^u\right\| \leq 2\lambda_h, \quad \text{where } c_3 \overset{\text{def}}{=} 2\lambda_h. \qquad (32)$$

Denoting $\sigma_j^2 \overset{\text{def}}{=} \mathbb{E}\left[(W_j^t)^2 \mid W_{j-1}^t, \ldots, W_1^t\right] = \mathbb{E}_{\xi_j}\left[(W_j^t)^2\right]$, Lemma 3 implies

$$\mathbb{P}\left\{|⑤| > b_3 \text{ and } \sum_{j=1}^t \bar{\sigma}_j^2 \leq G_3 \log\frac{2}{\delta''}\right\} \leq 2\exp\left(-\frac{b_3^2}{2G_3\log\frac{2}{\delta''} + \frac{2c_3b_3}{3}}\right) = \delta'',$$

where the last identity is true, if we set $b_3 = \left(\frac{1}{3}c_3 + \sqrt{\frac{1}{9}c_3 + 2G_3}\right)\log\frac{2}{\delta''}$. To define $G_3$, we need to bound $\sum_{j=1}^{t}\bar{\sigma}_j^2$:

$$
\begin{aligned}
\sum_{j=1}^{t}\bar{\sigma}_j^2 &= \sum_{j=1}^{t}\mathbb{E}_{q_j,\xi_j}\left[(W_j^t)^2\right] \overset{(32)}{\leq} \sum_{j=1}^{t}(1-\alpha)^{2(t-j+1)}\mathbb{E}_{q_j,\xi_j}\left[\|\omega_j^u\|^2\right] \\
&\overset{(25)}{\leq} \sum_{j=1}^{t}(1-\alpha)^{2(t-j+1)}\cdot 18\lambda_h^{2-p}\left(6L^p+3\sigma_h^p\right)\gamma^p \leq \frac{18\lambda_h^{2-p}(6L^p+3\sigma_h^p)\gamma^p}{1-(1-\alpha)^2} \\
&= \frac{18\lambda_h^{2-p}\left(6L^p+3\sigma_h^p\right)\gamma^p}{\alpha(2-\alpha)} \leq \frac{18\lambda_h^{2-p}\left(6L^p+3\sigma_h^p\right)\gamma^p}{\alpha} = G_3.
\end{aligned}
\tag{33}
$$

**Upper bound for ⑥.** The sequence $Z_1^t, \ldots, Z_t^t$ forms a martingale difference sequence, since the definition of $Z_j^t$ leads to $\mathbb{E}\left[Z_j^t \mid Z_{j-1}^t, \ldots, Z_1^t\right] = \mathbb{E}_{q_j,\xi_j}\left[Z_j^t\right] = 0$ for all $j \in [t]$. Also, according to Lemma 4, we obtain that

$$
|Z_j^t| \leq \left\|(1-\alpha)^{t-j+1}\omega_j^u\right\|^2 + \mathbb{E}_{q_j,\xi_j}\left[\left\|(1-\alpha)^{t-j+1}\omega_j^u\right\|^2\right] \leq 4\lambda_h^2 + 4\lambda_h^2 = 8\lambda_h^2,
\tag{34}
$$

where we define $c_4 \overset{\text{def}}{=} 8\lambda_h^2$. Denoting the conditional variance of $Z_j^t$ as $\widehat{\sigma}_j^2 \overset{\text{def}}{=} \mathbb{E}\left[(Z_j^t)^2 \mid Z_{j-1}^t, \ldots, Z_1^t\right] = \mathbb{E}_{q_j,\xi_j}\left[(Z_j^t)^2\right]$, we can bound $\widehat{\sigma}_j^2$ easily

$$
\begin{aligned}
\widehat{\sigma}_j^2 &\overset{(34)}{\leq} 8\lambda_h^2 \cdot \mathbb{E}_{q_j,\xi_j}\left[\left|\left\|(1-\alpha)^{t-j+1}\omega_j^u\right\|^2 - \mathbb{E}_{q_j,\xi_j}\left[\left\|(1-\alpha)^{t-j+1}\omega_j^u\right\|^2\right]\right|\right] \\
&\leq 16\lambda_h^2\mathbb{E}_{q_j,\xi_j}\left[\left\|(1-\alpha)^{t-j+1}\omega_j^u\right\|^2\right].
\end{aligned}
\tag{35}
$$

Then, Lemma 3 implies

$$
\mathbb{P}\left\{|⑥| > b_4 \text{ and } \sum_{j=1}^{t}\widehat{\sigma}_j^2 \leq G_4\log\frac{2}{\delta'}\right\} \leq 2\exp\left(-\frac{b_4^2}{2G_4\log\frac{2}{\delta'}+\frac{2c_4b_4}{3}}\right) = \delta'',
$$

where the last identity is true, if we set $b_4 = \left(\frac{1}{3}c_4 + \sqrt{\frac{1}{9}c_4 + 2G_4}\right)\log\frac{2}{\delta''}$. To define $G_4$, we need to bound $\sum_{j=1}^{t}\widehat{\sigma}_j^2$:

$$
\begin{aligned}
\sum_{j=1}^{t}\widehat{\sigma}_j^2 &\overset{(35)}{\leq} 16\lambda_h^2\sum_{j=1}^{t}\mathbb{E}_{q_j,\xi_j}\left[\left\|(1-\alpha)^{t-j+1}\omega_j^u\right\|^2\right] = 16\lambda_h^2\sum_{j=1}^{t}(1-\alpha)^{2(t-j+1)}\mathbb{E}_{q_j,\xi_j}\left[\left\|\omega_j^u\right\|^2\right] \\
&\leq 16\lambda_h^2 \cdot \frac{18\lambda_h^{2-p}\left(6L^p+3\sigma_h^p\right)\gamma^p}{\alpha} = \frac{16\cdot 18\lambda_h^{4-p}(6L^p+3\sigma_h^p)\gamma^p}{\alpha} = G_4.
\end{aligned}
$$

**Upper bound for ⑦.** Thanks to the proof of the bound on ④ (see (33)), we have already shown what we need: with probability 1

$$
⑦ = \sum_{j=1}^{t}\mathbb{E}_{\xi_j}\left[\left\|(1-\alpha)^{t-j+1}\omega_j^u\right\|^2\right] \overset{(33)}{\leq} \frac{18\lambda^{2-p}(6L^p+3\sigma^p)\gamma^p}{\alpha}.
$$

**Upper bound on ⑧.** By definition of $\omega_j^b$, with probability 1 we have

$$
\begin{aligned}
⑧ &\overset{(31)}{=} \left\|\sum_{j=1}^{t}(1-\alpha)^{t-j+1}\omega_j^b\right\| \leq \sum_{j=1}^{t}(1-\alpha)^{t-j+1}\|\omega_j^b\| \\
&\overset{(25)}{\leq} 4\lambda_h^{1-p}\left(6L^p+3\sigma_h^p\right)\gamma^p\sum_{j=1}^{t}(1-\alpha)^{t-j+1} \leq \frac{4\lambda_h^{1-p}\left(6L^p+3\sigma_h^p\right)\gamma^p}{\alpha}.
\end{aligned}
$$

To sum up, we introduce event $E_{⑤,t}$ as follows

$$E_{⑤,t} = \left\{ |⑤| \le b_3 \text{ or } \sum_{j=1}^{t} \bar{\sigma}_j^2 > G_3 \log \frac{2}{\delta''} \right\}, \tag{36}$$

where $c_3 = 2\lambda_h$, $G_3 = \frac{18\lambda_h^{2-p}(6L^p+3\sigma_h^p)\gamma^p}{\alpha}$, $b_3 = \left(\frac{1}{3}c_3 + \sqrt{\frac{1}{9}c_3^2 + 2G_3}\right) \log \frac{2}{\delta''}$. We have shown that $\mathbb{P}\{E_{⑤,t}\} \ge 1 - \delta''$. We adjust the bound on $b_3$:

$$
\begin{aligned}
b_3 &= \left(\frac{1}{3}c_3 + \sqrt{\frac{1}{9}c_3^2 + 2G_3}\right) \log \frac{2}{\delta'} \le \left(\frac{2}{3}c_3 + \sqrt{2G_3}\right) \log \frac{2}{\delta''} \\
&= \left(\frac{4}{3}\lambda_h + \sqrt{36\frac{\lambda_h^{2-p}\left(6L^p + 3\sigma_h^p\right)\gamma^p}{\alpha}}\right) \log \frac{2}{\delta''} \\
&= \lambda_h \left(\frac{4}{3} + 6\sqrt{\frac{(6L^p + 3\sigma_h^p)\gamma^p}{\alpha\lambda^p}}\right) \log \frac{2}{\delta''}.
\end{aligned}
$$

Also we introduce event $E_{⑥,t}$ as follows

$$E_{⑥,t} = \left\{ |⑥| \le b_4 \text{ or } \sum_{j=1}^{t} \widehat{\sigma}_j^2 > G_4 \log \frac{2}{\delta''} \right\}, \tag{37}$$

where $c_4 = 8\lambda_h^2$, $G_4 = \frac{16 \cdot 18\lambda_h^{4-p}\left(6L^p+3\sigma_h^p\right)\gamma^p}{\alpha}$, $b_4 = \left(\frac{1}{3}c_4 + \sqrt{\frac{1}{9}c_4^2 + 2G_4}\right)\log\frac{2}{\delta''}$. We have shown that $\mathbb{P}\{E_{⑥,t}\} \ge 1 - \delta''$. We adjust the bound on $b_4$:

$$
\begin{aligned}
b_4 &= \left(\frac{1}{3}c_4 + \sqrt{\frac{1}{9}c_4^2 + 2G_4}\right) \log \frac{2}{\delta'} \le \left(\frac{2}{3}c_4 + \sqrt{2G_4}\right) \log \frac{2}{\delta''} \\
&= \left(\frac{16}{3}\lambda_h^2 + \sqrt{\frac{16 \cdot 36\lambda_h^{4-p}\left(6L^p + 3\sigma_h^p\right)\gamma^p}{\alpha}}\right) \log \frac{2}{\delta''} \\
&= \lambda_h^2 \left(\frac{16}{3} + 24\sqrt{\frac{(6L^p + 3\sigma_h^p)\gamma^p}{\alpha\lambda_h^p}}\right) \log \frac{2}{\delta''}.
\end{aligned}
$$

Thus, the event $E_{⑤,t} \cap E_{⑥,t}$ implies

$$
\begin{aligned}
\left\| \sum_{j=1}^{t}(1-\alpha)^{t-j+1}\omega_j \right\| &\le \left\| \sum_{j=1}^{t}(1-\alpha)^{t-j+1}\omega_j^u \right\| + \left\| \sum_{j=1}^{t}(1-\alpha)^{t-j+1}\omega_j^b \right\| \\
&\le |⑤| + \sqrt{2 \cdot ⑥ + 2 \cdot ⑦} + ⑧ \\
&\le \lambda_h \left(\frac{4}{3} + 6\sqrt{\frac{(6L^p + 3\sigma_h^p)\gamma^p}{\alpha\lambda_h^p}}\right) \log \frac{2}{\delta''} + 4\lambda_h \cdot \frac{(6L^p + 3\sigma_h^p)\gamma^p}{\alpha\lambda_h^p} \\
&\quad + \sqrt{2\lambda_h^2 \left(\frac{16}{3} + 24\sqrt{\frac{(6L^p + 3\sigma_h^p)\gamma^p}{\alpha\lambda_h^p}}\right) \log \frac{2}{\delta''} + 36\lambda_h^2 \frac{(6L^p + 3\sigma_h^p)}{\alpha\lambda_h^p}}.
\end{aligned}
$$

Assuming that $\lambda_h \ge \frac{\gamma(L+\sigma_h)}{\alpha^{1/p}}$, we have the event $E_{⑤,t} \cap E_{⑥,t}$ implies

$$
\begin{aligned}
\left\| \sum_{j=1}^{t}(1-\alpha)^{t-j+1}\omega_j \right\| &\le \lambda_h \left(\frac{4}{3} + 6\sqrt{6} + 24 + \sqrt{\frac{32}{3} + 48\sqrt{6} + 216}\right) \log \frac{2}{\delta''} \\
&\le 59\lambda_h \log \frac{2}{\delta''}.
\end{aligned}
$$

$\square$

**Main Theorem** Now we are ready to state and prove the main results of this section: high-probability convergence guarantees for Algorithm 2.

**Theorem 5.** *Let Assumptions 1, 2, 3 and 4 hold, and $\Delta_1 = \Delta_0 = F(x_0) - F_*$ . Then if the parameters of Algorithm 2: stepsize*

$$\gamma \leq \min\left\{\frac{1}{2}\sqrt{\frac{\Delta_1}{LT}}, \frac{\alpha}{12}\sqrt{\frac{\Delta_1}{L}}, \frac{1}{1408\alpha T\log\frac{8T}{\delta}}\sqrt{\frac{\Delta_1}{L}}, \frac{\Delta_1}{352\sigma\alpha^{\frac{p-1}{p}}T\log\frac{8T}{\delta}}, \sqrt{\frac{\Delta_1\alpha^{1/p}}{968(L+\sigma_h)T\log\frac{8T}{\delta}}}\right\},$$
(38)

*momentum parameter and clipping levels*

$$\alpha = \frac{1}{T^{\frac{p}{2p-1}}}, \quad \lambda = \max\left\{4\sqrt{L\Delta_1}, \frac{\sigma}{\alpha^{1/p}}\right\}, \quad \lambda_h = \frac{2\gamma(L+\sigma_h)}{\alpha^{1/p}}$$
(39)

*for some $T \geq 1$ and $\delta \in (0,1]$ such that $\log\frac{8T}{\delta} \geq 1$. Then after $T$ iterations of Algorithm 2 the iterates with probability at least $1 - \delta$ satisfy*

$$\frac{1}{T}\sum_{t=0}^{T-1}\|\nabla F(x_t)\| \leq \frac{2\Delta_1}{\gamma T}.$$
(40)

*Moreover, if we set (38) as identity, then the iterates generated by Algorithm 2 after $T$ iterations with probability at least $1 - \delta$ satisfy*

$$\frac{1}{T}\sum_{t=0}^{T-1}\|\nabla F(x_t)\| = \mathcal{O}\left(\frac{\max\left\{\sqrt{\Delta_1(L+\sigma_h)}, \sigma\right\}}{T^{\frac{p-1}{2p-1}}}\log\frac{T}{\delta}\right),$$
(41)

*implying that to achieve $\frac{1}{T}\sum_{t=0}^{T-1}\|\nabla F(x_t)\| \leq \varepsilon$ with probability at least $1 - \delta$ Algorithm 2 requires*

$$T = \mathcal{O}\left(\left(\frac{\max\left\{\sqrt{\Delta_1(L+\sigma_h)}, \sigma\right\}}{\varepsilon}\right)^{\frac{2p-1}{p-1}}\log^{\frac{2p-1}{p-1}}\left(\frac{1}{\delta}\left(\frac{\max\left\{\sqrt{\Delta_1(L+\sigma_h)}, \sigma\right\}}{\varepsilon}\right)^{\frac{2p-1}{p-1}}\right)\right)$$
(42)

*iterations/samples.*

*Proof.* We remind in the proofs of Lemma 11 and Lemma 12 we have shown

$$E_{①,t} \stackrel{(29)}{=} \left\{|①| \leq b_1 \text{ or } \sum_{j=1}^{t}\sigma_j^2 > G_1\log\frac{2}{\delta'}\right\}, \qquad E_{②,t} \stackrel{(30)}{=} \left\{|②| \leq b_2 \text{ or } \sum_{j=1}^{t}\widetilde{\sigma}_j^2 > G_2\log\frac{2}{\delta'}\right\},$$

$$E_{⑤,t} \stackrel{(36)}{=} \left\{|⑤| \leq b_3 \text{ or } \sum_{j=1}^{t}\bar{\sigma}_j^2 > G_3\log\frac{2}{\delta''}\right\}, \qquad E_{⑥,t} \stackrel{(37)}{=} \left\{|⑥| \leq b_4 \text{ or } \sum_{j=1}^{t}\widehat{\sigma}_j^2 > G_4\log\frac{2}{\delta''}\right\},$$

where

$$c_1 = 2\lambda, \ G_1 = \frac{18\lambda^{2-p}\sigma^p}{\alpha}, \ b_1 \leq \lambda\left(\frac{4}{3} + 6\sqrt{\frac{1}{\alpha}\left(\frac{\sigma}{\lambda}\right)^p}\right)\log\frac{2}{\delta'},$$

$$c_2 = 8\lambda^2, \ G_2 = \frac{16\cdot18\lambda^{4-p}\sigma^p}{\alpha}, \ b_2 \leq \lambda^2\left(\frac{16}{3} + 24\sqrt{\frac{1}{\alpha}\left(\frac{\sigma}{\lambda}\right)^p}\right)\log\frac{2}{\delta'},$$

$$c_3 = 2\lambda_h, \ G_3 = \frac{18\lambda_h^{2-p}\left(6L^p + 3\sigma_h^p\right)\gamma^p}{\alpha}, \ b_3 \leq \lambda_h\left(\frac{4}{3} + 6\sqrt{\frac{(6L^p + 3\sigma_h^p)\gamma^p}{\alpha\lambda^p}}\right)\log\frac{2}{\delta''},$$

$$c_4 = 8\lambda_h^2, \ G_4 = \frac{16\cdot18\lambda_h^{4-p}\left(6L^p + 3\sigma_h^p\right)\gamma^p}{\alpha}, \ b_4 \leq \lambda_h^2\left(\frac{16}{3} + 24\sqrt{\frac{(6L^p + 3\sigma_h^p)\gamma^p}{\alpha\lambda_h^p}}\right)\log\frac{2}{\delta''}.$$

Moreover, according to Lemma 11 and Lemma 12, we have

$$\mathbb{P}\left\{E_{①,t}\right\} \geq 1 - \delta', \ \ \mathbb{P}\left\{E_{②,t}\right\} \geq 1 - \delta', \ \ \mathbb{P}\left\{E_{⑤,t}\right\} \geq 1 - \delta'', \ \ \mathbb{P}\left\{E_{⑥,t}\right\} \geq 1 - \delta''.$$

The idea of the proof is based on technique form work of Gorbunov et al. [2020], Sadiev et al. [2023], Liu et al. [2023b]: via mathematical induction we plan to show that for any $\tau \in \{0, 1, \ldots, T-1\}$ the event

$$G_\tau = E_\tau \cap E_{1,\tau} \cap E_{2,\tau} \cap E_{3,\tau} \cap E_{4,\tau}$$

holds with probability at least $1 - \frac{\tau\delta}{T}$, where event $E_\tau$ is defined as

$$E_\tau = \left\{ \gamma \sum_{s=0}^{t} \|\nabla F(x_s)\| + \Delta_{t+1} \le 2\Delta_1, \quad \forall\, t \le \tau \right\},$$

and

$$E_{1,\tau} = \bigcap_{t=1}^{\tau} E_{①,t}, \quad E_{2,\tau} = \bigcap_{t=1}^{\tau} E_{②,t}, \quad E_{3,\tau} = \bigcap_{t=1}^{\tau} E_{⑤,t}, \quad E_{4,\tau} = \bigcap_{t=1}^{\tau} E_{⑥,t}.$$

**Basis of induction.** Obviously, we have $G_0 = E_0 = \{\Delta_0 = \Delta_1 \le 2\Delta_1\}$ holds with probability $1 - \frac{\tau\delta}{T} \overset{\tau=0}{=} 1$.

**Step of induction.** Assume that the induction hypothesis is true for $\tau - 1$: $\mathbb{P}\{G_{\tau-1}\} \ge 1 - (\tau-1)\delta/T$. One needs to prove $\mathbb{P}\{G_\tau\} \ge 1 - \tau\delta/T$. We want to mention that

$$E_{\tau-1} \cap E_{1,\tau} \cap E_{2,\tau} \cap E_{3,\tau} \cap E_{4,\tau} = G_{\tau-1} \cap E_{①,\tau} \cap E_{②,\tau} \cap E_{⑤,\tau} \cap E_{⑥,\tau}.$$

Then, we have

$$
\begin{aligned}
\mathbb{P}\{E_{\tau-1} \cap E_{1,\tau} \cap E_{2,\tau} \cap E_{3,\tau} \cap E_{4,\tau}\} &= \mathbb{P}\{G_{\tau-1} \cap E_{①,\tau} \cap E_{②,\tau} \cap E_{⑤,\tau} \cap E_{⑥,\tau}\} \\
&= 1 - \mathbb{P}\{\overline{G_{\tau-1} \cap E_{①,\tau} \cap E_{②,\tau} \cap E_{⑤,\tau} \cap E_{⑥,\tau}}\} \\
&\ge 1 - \mathbb{P}\{\overline{G_{\tau-1}}\} - \mathbb{P}\{\overline{E_{①,\tau}}\} - \mathbb{P}\{\overline{E_{②,\tau}}\} \\
&\quad - \mathbb{P}\{\overline{E_{⑤,\tau}}\} - \mathbb{P}\{\overline{E_{⑥,\tau}}\} \\
&\ge 1 - \frac{(\tau-1)\delta}{T} - 2\delta' - 2\delta'' \\
&= 1 - \frac{\tau\delta}{T} + \left(\frac{\delta}{T} - 2\delta' - 2\delta''\right) \\
&= 1 - \frac{\tau\delta}{T},
\end{aligned}
$$

where we have selected $\delta' = \delta'' = \frac{\delta}{4T}$. The event $E_{\tau-1}$ implies for any $t \le \tau$

$$\Delta_t \le \gamma \sum_{s=0}^{t-1} \|\nabla F(x_s)\| + \Delta_t \le 2\Delta_1.$$

Therefore, we have $\|\nabla F(x_t)\| \le \sqrt{2L\Delta_t} \le 2\sqrt{L\Delta_1} \le \frac{\lambda}{2}$. Assuming $E_{\tau-1} \cap E_{1,\tau} \cap E_{2,\tau} \cap E_{3,\tau} \cap E_{4,\tau}$ happens, Lemma 10 implies

$$
\begin{aligned}
\gamma \sum_{t=0}^{\tau} \|\nabla F(x_t)\| + \Delta_{\tau+1} &\le \Delta_1 + \frac{3\gamma}{\alpha}\sqrt{L\Delta_1} + \frac{\gamma^2 L(\tau+1)}{2} \\
&\quad + 2\gamma\alpha \sum_{t=1}^{\tau} \left\|\sum_{j=1}^{t}(1-\alpha)^{t-j}\theta_j\right\| + 2\gamma \sum_{t=1}^{\tau}\left\|\sum_{j=1}^{t}(1-\alpha)^{t-j+1}\omega_j\right\| \\
&\le \Delta_1 + \frac{3\gamma}{\alpha}\sqrt{L\Delta_1} + \frac{\gamma^2 L(\tau+1)}{2} \\
&\quad + 44\gamma\alpha(\tau+1)\lambda \log\frac{8T}{\delta} + 118\gamma(\tau+1)\lambda_h \log\frac{8T}{\delta}.
\end{aligned}
$$

By setting clipping levels (39) and stepsize (38), we have

$$\gamma \sum_{t=0}^{\tau} \|\nabla F(x_t)\| + \Delta_{\tau+1} \le \Delta_1 + \frac{\Delta_1}{4} + \frac{\Delta_1}{4} + \frac{\Delta_1}{4} + \frac{\Delta_1}{4} = 2\Delta_1.$$

Therefore, we obtain

$$\begin{aligned}
\mathbb{P}\left\{G_\tau\right\} &= \mathbb{P}\left\{E_\tau \cap E_{1,\tau} \cap E_{2,\tau} \cap E_{3,\tau} \cap E_{4,\tau}\right\} \\
&= \mathbb{P}\left\{E_{\tau-1} \cap E_{1,\tau} \cap E_{2,\tau} \cap E_{3,\tau} \cap E_{4,\tau}\right\} \\
&\geq 1 - \frac{\tau\delta}{T}.
\end{aligned}$$

Thus, we have

$$\mathbb{P}\left\{E_T\right\} \geq \mathbb{P}\left\{G_T\right\} \geq 1 - \delta,$$

or equivalently with probability at least $1 - \delta$

$$\gamma \sum_{t=0}^{T-1} \|\nabla F(x_t)\| + \Delta_T \leq 2\Delta_1 \;\Rightarrow\; \frac{1}{T}\sum_{t=0}^{T-1} \|\nabla F(x_t)\| \leq \frac{2\Delta_1}{\gamma T}.$$

Pugging (38) into the inequality above, we have

$$\frac{1}{T}\sum_{t=0}^{T-1} \|\nabla F(x_t)\| \leq \frac{2\Delta_1}{\gamma T}$$

$$= \mathcal{O}\left(\max\left\{\sqrt{\frac{L\Delta_1}{T}}, \frac{\sqrt{L\Delta_1}}{\alpha T}, \alpha\sqrt{L\Delta_1}\log\frac{T}{\delta}, \sigma\alpha^{\frac{p-1}{p}}\log\frac{T}{\delta}, \sqrt{\frac{\Delta_1(L+\sigma_h)\log\frac{T}{\delta}}{T\alpha^{1/p}}}\right\}\right).$$

Selecting $\alpha = 1/T^{-p/2p-1}$ (see (39)), we obtain

$$\frac{1}{T}\sum_{t=1}^{T} \|\nabla F(x_t)\|$$

$$= \mathcal{O}\left(\max\left\{\sqrt{\frac{L\Delta_1}{T}}, \frac{\sqrt{L\Delta_1}}{T^{\frac{p-1}{2p-1}}}, \frac{\sqrt{L\Delta_1}}{T^{\frac{p-1}{2p-1}}}\log\frac{T}{\delta}, \frac{\sigma}{T^{\frac{p-1}{2p-1}}}\log\frac{T}{\delta}, \sqrt{\frac{\Delta_1(L+\sigma_h)\log\frac{T}{\delta}}{T^{2p-2/2p-1}}}\right\}\right)$$

$$= \mathcal{O}\left(\frac{\max\left\{\sqrt{\Delta_1(L+\sigma_h)}, \sigma\right\}}{T^{\frac{p-1}{2p-1}}}\log\frac{T}{\delta}\right).$$

**Discussion.** According to Theorem 3, the sample complexity of Algorithm 2 is

$$\widetilde{\mathcal{O}}\left(\left(\frac{\max\left\{\sqrt{\Delta_1(L+\sigma_h)}, \sigma\right\}}{\varepsilon}\right)^{\frac{2p-1}{p-1}}\right),$$

which has the same dependence on $\varepsilon$ as lower bounds do (see Theorem 1). However, the obtained upper bound has worst dependence on parameters $L, \sigma, \sigma_h, \Delta_1$ than lower bounds have. We suppose that this caused by the analysis technique, since Algorithm 1 has better convergence rate in terms of parameter dependence. This observation raises an interesting question: Is it possible to conduct high-probability analysis for Algorithm 1, or equivalently, Algorithm 2 without clipping?

$\square$

# G  Experimental Results

We solve a simple quadratic problem, $\frac{1}{2}\|x\|^2$, in dimension $d = 10$ , with synthetic noise sampled from a two-sided Pareto distribution. Recall that two-sided Pareto distribution with tail index $\bar{p}$ satisfies Assumption 3 assumption with $p < \bar{p}$. We consider different values of the tail index $\bar{p}$ ranging from 1.1 to 2.0. We start each algorithm from the same point sampled uniformly at random from the standard normal distribution. Our empirical analysis consists of three experimental settings:

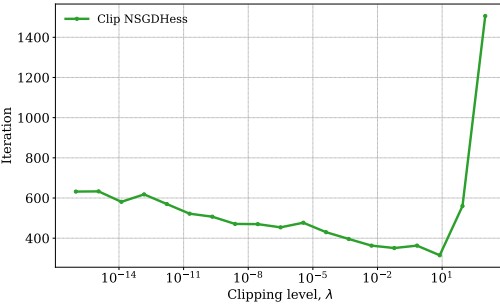

Figure 3: Effect of Hessian Clipping Level $\lambda_h = \lambda$ on the Iteration Complexity. The plot shows the number of iterations required for Clip NSGDHess to find a point with $\|\nabla F(x)\| \leq 3/2$. For extremely small and large values of $\lambda$, more iterations are needed. The recommended value for this task is $\lambda_h = 10$.

Figure 4: Number of iterations needed for Clip NSGDMHess and Clip NSGDM under Varying Tail Index to find a point with $\|\nabla F(x)\| \leq 3/2$ starting with the same initial point. The performance of both algorithms decreases gradually with the decrease of the tail index. The iteration complexity of the second-order algorithm, Clip NSGDMHess is uniformly better for all values of $p \in [1.1, 2]$.

**1. Effect of Clipping Level on Iteration Complexity.**   We run Algorithm 2 (Clip NSGDHess) with $\lambda_h = \lambda$ until it reached the target accuracy $\|\nabla F(x)\| \leq 3/2$. The stepsize and momentum parameters are set to $\gamma = 0.01$ and $\alpha = 0.2$. Clipping levels are varied from $10^{-16}$ to $10^3$ in multiplicative steps of 10. In Figure 3, we observe that intermediate clipping ($\lambda = 10$) yields the lowest iteration complexity, also refer to Table 2 for precise numerical values. Extremely small or large values lead to slower convergence. The proposed algorithm tolerates very small values of $\lambda$, but very large values (e.g., $\lambda > 10^3$) result in very slow convergence. This empirically confirms the need for Hessian clipping to stabilize convergence and achieve fast convergence.

| Clipping Level, $\lambda$ | $10^{-16}$ | $10^{-8}$ | $10^0$ | $10^2$ | $10^3$ |
|---|---|---|---|---|---|
| Iterations | 632 | 471 | 315 | 560 | 1506 |

Table 2: Iteration complexity depending on the clipping level $\lambda = \lambda_h$ for Algorithm 2 (Clip NSGDHess), cf. Figure 4.

**2. Comparison with Clip NSGDM under Varying Tail Index.**   We compare Clip NSGDHess with Clip NSGDM (which was proposed in Cutkosky and Mehta [2021]). For Clip NSGDM, we used the theoretical parameter choices: stepsize $\gamma = T^{-\frac{2p-1}{3p-2}}$ and momentum $\alpha = T^{-\frac{p}{3p-2}}$. For both methods we fixed $T = 4000$, $\lambda = 0.5$, and $\bar{\lambda}_h = 0.05$. These experiments were conducted to complement Figure 1, which presents the theoretical iteration complexities. As shown in Figure 4, the empirical results align with the theory: Clip NSGDHess consistently outperforms Clip NSGDM, and the performance gap becomes larger as $p$ increases. This provides strong empirical support for our theoretical findings.

**3. Sensitivity to Gradient Clipping for Fixed Hessian Clipping.**   In this set of experiments, we test Clip NSGDHess (Algorithm 2). Our goal is to study how iteration complexity depends on the choice of the gradient clipping level $\lambda$ when the Hessian clipping level is fixed. Specifically, we test three different values of the Hessian clipping level, and vary the gradient clipping threshold $\lambda$:

- For $\bar{\lambda}_h = 0.01$: $\lambda$ ranged from $10^{-4}$ to $10^2$ (step 10).
- For $\bar{\lambda}_h = 1.0$: $\lambda$ ranged from $10^{-2}$ to $10^4$ (step 10).
- For $\bar{\lambda}_h = 10$: $\lambda$ ranged from $10^0$ to $10^6$ (step 10).

The results are summarized in Figure 5.

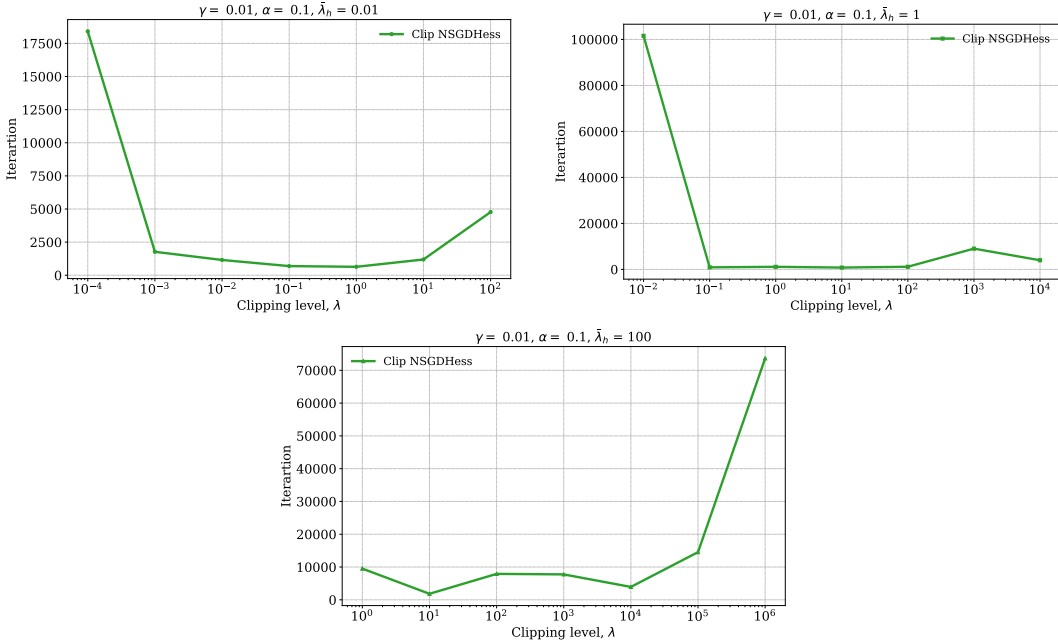

Figure 5: Iteration complexity of Clip NSGDHess (Algorithm 2) depending on gradient clipping for the three different fixed values of Hessian clipping $\bar{\lambda}_h \in \{0.01, 1, 100\}$.

In all cases, the algorithm is tested until $\|\nabla F(x)\| \leq \frac{1}{2}$. We observe that intermediate values of $\lambda$ yield the best performance across all three regimes (small, medium, and large Hessian clipping levels). As shown in Figure 5, both extremely small and extremely large values of $\lambda$ lead to higher iteration complexity, while moderate choices minimize it. The optimal values are:

$$\lambda = 1.0 \quad \text{for} \quad \bar{\lambda}_h = 0.01, \qquad \lambda = 0.1 \quad \text{for} \quad \bar{\lambda}_h = 1.0, \qquad \lambda = 10.0 \quad \text{for} \quad \bar{\lambda}_h = 100.0.$$

These findings confirm that moderate clipping levels are crucial for achieving the best performance aligning with our theoretical findings in Theorem 3.

