# OpenReview forum: "Second-order Optimization under Heavy-Tailed Noise: Hessian Clipping and Sample Complexity Limits"
_NeurIPS.cc/2025/Conference — NeurIPS 2025 poster_

### Official Review · Reviewer_tAtg · 2025-07-01

**Clarity:** 4
**Significance:** 4
**Originality:** 3
**Rating:** 5
**Confidence:** 4

**Summary:**

This paper studies the problem of second-order stochastic optimization (SOSO) under heavy-tailed noise, a setting where both stochastic gradients and Hessians only have bounded p-th moments, which is common in modern machine learning tasks with outliers or data heterogeneity. The paper establishes tight minimax lower bounds on the sample complexity for any second-order optimization algorithm under heavy-tailed noise. The results demonstrate that second-order methods can still achieve provable improvements over first-order methods, even in heavy-tailed regimes. The paper proposes a normalized SGD with Hessian correction algorithm that incorporates second-order information and provably achieves the established lower bond for finding approximate stationary points. The paper also introduces Hessian clipping, an extension of gradient clipping to handle extreme noise in Hessian estimates.

**Questions:**

Please refer the the weaknesses section.

**Ethical Concerns:**

["NO or VERY MINOR ethics concerns only"]

**Quality:**

4

**Strengths And Weaknesses:**

Strengths:
- This paper is well-written and well-structured, easy to read and follow.
- This paper provides rigorous theoretical contribution.
- The analysis is technically sound and comprehensive, covering both in-expectation and high-probability results and addressing the impact of gradient and Hessian noise.
- The originality of this paper is clear and solid.
Weaknesses:
- The experimental validation is minimal, consisting only of toy synthetic experiments in low dimensions. There is no evaluation on real-world machine learning tasks to demonstrate practical effectiveness.
- The proposed methods require assumptions on hyperparameters and question setup, which may limit their practical usability.
- While theory is strong, there is no discussion of computational costs for Hessian-vector products or clipping in large-scale models.

---

> ### Author Rebuttal · Authors · 2025-07-30
>
> Dear Reviewer tAtg,
>
> Thank you for your thorough and thoughtful review, and for your positive evaluation of the paper’s clarity, originality, and theoretical contribution. We are especially grateful for your recognition of the strength of our analysis, including both in-expectation and high-probability results under heavy-tailed noise. Below, we respond to your comments regarding experimental validation, practical considerations, and assumptions, and we appreciate your suggestions for further strengthening the paper.
>
>
> > __Q1:__ The experimental validation is minimal, consisting only of toy synthetic experiments in low dimensions. There is no evaluation on real-world machine learning tasks to demonstrate practical effectiveness.
>
>
> __A1:__ We acknowledge the importance of empirical evaluation of the algorithms on real-world machine learning tasks, and we are planning to test our algorithms on more serious tasks in the future. However, the main goal of this work is to understand the theoretical complexity limits of second-order stochastic algorithms under challenging heavy-tailed settings. Our Algorithm 1 was already tested before in reinforcement learning literature [Salehkaleybar et al., 2022, Fatkhullin et al., 2023] with promising results. We are hopeful that our theoretical enhancement with gradient and Hessian clipping (Algorithm 2) will further improve the practical performance, and we leave this for future work.
>
> We also find controlled synthetic experiments insightful and important, especially in the initial phase of algorithmic development. We conduct the following additional ablation studies, which provide insights on theoretically derived complexities and the sensitivity to clipping threshold tuning.
>
> We solved a simple quadratic problem, $\frac{1}{2}\|x\|^2$, in dimension $d = 10$, with synthetic noise sampled from a symmetrized Pareto distribution. We considered both a fixed heavy-tailed regime with tail index $p = 1.1$ and a variable tail index ranging from $1.1$ to $2.0$. We start each algorithm from the same point sampled uniformly at random from the standard normal distribution.
> Our empirical analysis consists of three experimental settings:
>
> __1. Effect of Clipping Level on Iteration Complexity:__
> We ran Algorithm 2 (Clip NSGDHess) with $\lambda_h = \lambda$ until it reached the target accuracy of $\|\nabla F(x)\| \leq 3/2$. The stepsize and momentum parameters were set to $\gamma = 0.01$ and $\alpha = 0.2$. Clipping levels were varied from $10^{-16}$ to $10^3$ using a multiplicative step of 10. The results are shown below:
> | $\lambda$        | $10^{-16}$ | $10^{-8}$ | $10$ | $10^{2}$ | $10^{3}$ |
> |------------------|------------|-----------|------|----------|----------|
> | Iterations       | $632$        | $471$       | $\mathbf{321}$ | $562$      | $1506$     |
>
> __Table 1:__ Intermediate clipping ($\lambda = 10$) yields the lowest iteration complexity. Extremely small or large values lead to slower convergence. The proposed algorithm can gracefully tolerate very small values of clipping parameter $\lambda$, but using values larger than $\lambda = 10^3$ leads to very slow convergence. This observation motivates the need for gradient and hessian clipping.
>
> __2.Comparison with Clip NSGDM under Varying Tail Index:__
> We compared Clip NSGDHess with Clip NSGDM (as proposed in [2]). For Clip NSGDM, we used the stepsize $\gamma = T^{-\frac{2p-1}{3p-2}}$ and momentum $\alpha = T^{-\frac{p}{3p-2}}$ according to theory. For Clip NSGDHess, both parameters were set to $T^{-\frac{p}{2p-1}}$. We fixed $\lambda = 0.5$, $\bar{\lambda}_h = 0.05$, and $T = 4000$ iterations.
>
> | Tail index, $p$ | Iteration Complexity (Clip NSGDM) | Iteration Complexity (Clip NSGDHess) |
> |----------------|--------------------------------------------|--------------------------------------|
> | $1.2$            | $2476$                                       | $2128$                                 |
> | $1.6$            | $1183$                                       | $726 $                                 |
> | $2.0$            | $877$                                        | $444$                                  |
>
>
> __Table 2:__ Clip NSGDHess consistently outperforms Clip NSGDM, with the gap increasing for smaller $p$, i.e., heavier-tailed noise. Visually the picture looks similar to theoretical complexities in Figure 1 of our original submission.
>
> [2] Cutkosky, Ashok, and Harsh Mehta. "High-probability bounds for non-convex stochastic optimization with heavy tails." Advances in Neural Information Processing Systems 34 (2021): 4883-4895.
>
>
> __3.Sensitivity to Gradient Clipping for Fixed Hessian Clipping:__
> We fixed the Hessian clipping level and varied the gradient clipping level $\lambda$:
>
> - For $\bar{\lambda}_{h} = 0.1$: we set $\lambda$ from $10^{-3}$ to $10^{3}$ with multiplicative step $10$;
> - For $\bar{\lambda}{h} = 1.0$: we set  $\lambda$ from $10^{-2}$ to $10^{4}$ with multiplicative step $10$;
> - For $\bar{\lambda}{h} = 10$:  we set  $\lambda$ from $10^{-1}$ to $10^{5}$ with multiplicative step $10$.
>
> In all cases, the algorithm was run until $\|\nabla F(x)\| \leq \frac{1}{2}$. We consistently observed that both extremely small and large values of $\lambda$ led to higher iteration complexity. The optimal values were:
>
> - $\lambda = 0.01$ for $\bar{\lambda}_h = 0.1$;
> - $\lambda = 0.1$ for $\bar{\lambda}_h = 1.0$;
> - $\lambda = 1.0$ for $\bar{\lambda}_h = 10$.
>
> This pattern confirms that intermediate clipping levels are crucial for performance.
>
>
> > __Q2:__ The proposed methods require assumptions on hyperparameters and question setup, which may limit their practical usability.
>
> __A2:__ This is a valid concern, and we can offer a strategy that avoids any tuning requirement in Algorithm 1. We can set step-size $\gamma = \sqrt{\frac{ \alpha^{\frac{1}{2}}}{ T}}$, momentum $\alpha = \left(\frac{1}{T}\right)^{\frac{2}{3}}$. Following the proof strategy similar to (Hübler et al, 2024), we can establish convergence of our algorithm with this step-size strategy. Similarly to the observations in (Hübler et al, 2024), our convergence rate will become slower: $\mathcal{O}(T^{-\frac{p-1}{3p}})$ instead of $\mathcal{O}(T^{-\frac{p-1}{2p-1}})$ (and the constant behind $\mathcal{O}$) will be larger. Nevertheless, this strategy avoids any parameter tuning and ensures non-asymptotic convergence. In case of Algorithm 2, the situation is more complex and we do not have a good strategy for setting clipping thresholds when problem parameters are unknown. Since the main focus of our paper is understanding the sample complexity upper and lower bounds when all parameters are known, we omit the discussion of these details.
>
> Florian Hübler, Ilyas Fatkhullin, Niao He. From Gradient Clipping to Normalization for Heavy Tailed SGD. AISTATS 2025.
>
>
> > __Q3:__ While theory is strong, there is no discussion of computational costs for Hessian-vector products or clipping in large-scale models.
>
> __A3:__ Thank you for your comment! The computational costs for Hessian-vector products is almost the same as the costs for gradients, refer to e.g., to (Pearlmutter, 1994). The idea is that let us imagine we have a backpropagation procedure for computing the gradient of $f(x)$. After this procedure we get $\nabla f(x)$. To compute Hessian-vector product $\nabla^2 f(x) v$ we need to construct a function $\langle \nabla f(x), v\rangle$, and compute the gradient of the new function with respect to $x$ via backpropagation Therefore, we obtain the desired Hessian-vector $\nabla^2 f(x) v$. Thus, the computational cost is $\mathcal O(d)$.
>
>  Pearlmutter, Barak A. "Fast exact multiplication by the Hessian." Neural computation 6, no. 1 (1994): 147-160.

---

> > ### Comment · Reviewer_tAtg · 2025-08-05
> >
> > Thank you for your detailed response to my review. I appreciate you taking the time to address the points raised.

---

### Official Review · Reviewer_JJgW · 2025-07-02

**Clarity:** 2
**Significance:** 3
**Originality:** 2
**Rating:** 4
**Confidence:** 3

**Summary:**

This paper studies second-order optimization algorithms with heavy-tailed noise. A lower bound on the sample complexity for second-order algorithms is established along with an algorithm to match it in terms of first moment error. A second algorithm with clipping with high probability guarantees for the estimation error is proposed and shown to achieve near-optimal sample complexity.

**Questions:**

- Is \xi assumed to be i.i.d.? If not, is the expectation taken at steady-state (if \xi is Markovian for example)?The authors need to explicitly state their assumptions on the noise.

- Is the BCM assumption that much weaker than the p-moment bound assumption of [Zhang et al., 2020]? Why is the difference between these assumptions that important?

-What is the difference between O(.) and \Omega(.)? Why the need for two different notations? Also, Thm 1 uses \Theta(.), what is that?

**Ethical Concerns:**

["NO or VERY MINOR ethics concerns only"]

**Final Justification:**

The authors' rebuttal addressed my concerns. I retain my initial score, which recommended acceptance.

**Limitations:**

The assumptions of this paper are clearly identified and the authors provide a discussion on thee limitations of their work.

**Paper Formatting Concerns:**

I have no concerns on this matter

**Quality:**

2

**Strengths And Weaknesses:**

This paper has interesting contributions which, as far as I am concerned, are indeed novel. The assumptions, contributions and setting of the paper are clearly identified early on, making it easy to read. I was not able to check the proofs in detail.

Regardless, I would encourage the authors to proof-read their paper as I found a few typos while reading it. (e.g. E[  || \nabla f(x,\xi) || ]^p \leq B when talking about bounds on pth moment).

One weakness of this submission is the lack of numerical studies to illustrate the theory developed in the paper. While, the authors provide two plots in page 3, very little discussion is provided regarding the setup of their experiments and how these plots are related to the theorems in their paper. I would encourage the authors to include a simple toy problem illustrating their sample complexity bound. Figure 1 shows a plot of the sample complexity, but not details on this experiment are provided.

In terms of clarity, I also think that there is some work needed. For example, the authors should explicit say which algorithm (i.e. first equation of section B) they focus on in the very beginning of the paper to engage the reader and avoid confusion. Also, the authors reference Table 1 in the main text, but do not mention this table is in the supplementary material, which might create confusion.

I think a statement on how the choice of g_0 changes the sample complexity of algorithm 2 should be included in the main text. Fine details do not need to be included, but the fact that it only changes a constant on the sample complexity and not the dependency on \epsilon.

---

> ### Author Rebuttal · Authors · 2025-07-30
>
> Dear Reviewer JJgW,
>
> Thank you for your review and for your positive evaluation of the novelty and clarity of our contributions. We are glad to hear that you found the setting, assumptions, and theoretical results clearly presented. We also appreciate your constructive suggestions regarding clarity, notation, and experimental presentation, and we address each of your comments and questions in detail below.
>
>
> > __Q1:__ I would encourage the authors to include a simple toy problem illustrating their sample complexity bound.
>
> __A1:__ We want to mention that Figure 1 is not an experiment. It is an illustration of theoretically optimal complexity bounds (upper and lower bounds are matching) for SOSO and FOSO in order to support understanding. This plot implies that for the entire range of tail indices $p \in (1, 2]$, methods with access to second-order information have reduced sample complexity than methods that only have access to first-order information. Figure 2 is a toy experiment, which motivates us to study second-order methods and algorithms with clipping in depth. The most important information related to this experiment is provided in the caption to Figure 2. During the rebuttal phase, we conduct the following additional ablation studies on the same toy problem, which provide insights on theoretically derived complexities and the sensitivity to clipping threshold tuning.
>
> We solved a simple quadratic problem, $\frac{1}{2}\|x\|^2$, in dimension $d = 10$, with synthetic noise sampled from a symmetrized Pareto distribution. We considered both a fixed heavy-tailed regime with tail index $p = 1.1$ and a variable tail index ranging from $1.1$ to $2.0$. We start each algorithm from the same point sampled uniformly at random from the standard normal distribution.
> Our empirical analysis consists of three experimental settings:
>
> __1. Effect of Clipping Level on Iteration Complexity:__
> We ran Algorithm 2 (Clip NSGDHess) with $\lambda_h = \lambda$ until it reached the target accuracy of $\|\nabla F(x)\| \leq 3/2$. The stepsize and momentum parameters were set to $\gamma = 0.01$ and $\alpha = 0.2$. Clipping levels were varied from $10^{-16}$ to $10^3$ using a multiplicative step of 10. The results are shown below:
> | $\lambda$        | $10^{-16}$ | $10^{-8}$ | $10$ | $10^{2}$ | $10^{3}$ |
> |------------------|------------|-----------|------|----------|----------|
> | Iterations       | $632$        | $471$       | $\mathbf{321}$ | $562$      | $1506$     |
>
> __Table 1:__ Intermediate clipping ($\lambda = 10$) yields the lowest iteration complexity. Extremely small or large values lead to slower convergence. The proposed algorithm can gracefully tolerate very small values of clipping parameter $\lambda$, but using values larger than $\lambda = 10^3$ leads to very slow convergence. This observation motivates the need for gradient and hessian clipping.
>
> __2.Comparison with Clip NSGDM under Varying Tail Index:__
> We compared Clip NSGDHess with Clip NSGDM (as proposed in [2]). For Clip NSGDM, we used the stepsize $\gamma = T^{-\frac{2p-1}{3p-2}}$ and momentum $\alpha = T^{-\frac{p}{3p-2}}$ according to theory. For Clip NSGDHess, both parameters were set to $T^{-\frac{p}{2p-1}}$. We fixed $\lambda = 0.5$, $\bar{\lambda}_h = 0.05$, and $T = 4000$ iterations.
>
> | Tail index, $p$ | Iteration Complexity (Clip NSGDM) | Iteration Complexity (Clip NSGDHess) |
> |----------------|--------------------------------------------|--------------------------------------|
> | $1.2$            | $2476$                                       | $2128$                                 |
> | $1.6$            | $1183$                                       | $726 $                                 |
> | $2.0$            | $877$                                        | $444$                                  |
>
>
> __Table 2:__ Clip NSGDHess consistently outperforms Clip NSGDM, with the gap increasing for smaller $p$, i.e., heavier-tailed noise. Visually the picture looks similar to theoretical complexities in Figure 1 of our original submission.
>
> [2] Cutkosky, Ashok, and Harsh Mehta. "High-probability bounds for non-convex stochastic optimization with heavy tails." Advances in Neural Information Processing Systems 34 (2021): 4883-4895.
>
>
> __3.Sensitivity to Gradient Clipping for Fixed Hessian Clipping:__
> We fixed the Hessian clipping level and varied the gradient clipping level $\lambda$:
>
> - For $\bar{\lambda}_{h} = 0.1$: we set $\lambda$ from $10^{-3}$ to $10^{3}$ with multiplicative step $10$;
> - For $\bar{\lambda}{h} = 1.0$: we set  $\lambda$ from $10^{-2}$ to $10^{4}$ with multiplicative step $10$;
> - For $\bar{\lambda}{h} = 10$:  we set  $\lambda$ from $10^{-1}$ to $10^{5}$ with multiplicative step $10$.
>
> In all cases, the algorithm was run until $\|\nabla F(x)\| \leq \frac{1}{2}$. We consistently observed that both extremely small and large values of $\lambda$ led to higher iteration complexity. The optimal values were:
>
> - $\lambda = 0.01$ for $\bar{\lambda}_h = 0.1$;
> - $\lambda = 0.1$ for $\bar{\lambda}_h = 1.0$;
> - $\lambda = 1.0$ for $\bar{\lambda}_h = 10$.
>
> This pattern confirms that intermediate clipping levels are crucial for performance.
>
>
>
> > __Q2:__ I think a statement on how the choice of g_0 changes the sample complexity of algorithm 2 should be included in the main text. Fine details do not need to be included, but the fact that it only changes a constant on the sample complexity and not the dependency on \epsilon.
>
> __A2:__ Thank you for the suggestion! We will add an additional discussion about it in the main text of the camera-ready version.
>
> >  __Q3:__  Is \xi assumed to be i.i.d.? If not, is the expectation taken at steady-state (if \xi is Markovian for example)? The authors need to explicitly state their assumptions on the noise.
>
> __A3:__ On each iteration we sample $\xi$ independently from the same distribution, so the sequence $(\xi_t)_{t \geq 1}$ is assumed to be i.i.d. We will clarify this in the revision.
>
> >  __Q4:__  Is the BCM assumption that much weaker than the p-moment bound assumption of [Zhang et al., 2020]? Why is the difference between these assumptions that important?
>
> __A4:__ Our $p$-BCM assumption is weaker than $p$-th bounded moment. To see this, it is sufficient to show it for $p=2$. By bias-variance decomposition
> $\mathbb E \|\nabla f(x, \xi)\|^2 = \|\nabla F(x)\|^2 + \mathbb E \|\xi\|^2 \geq \mathbb E \|\xi\|^2. $
> On the other hand, $p$-th bounded moment assumption can be restrictive as it implies bounded gradients even in case $p=2$ as can be seen from the above decomposition. The class of Lipschitz smooth functions with unbounded gradients is much richer than the one with bounded gradients. In particular, it includes an important sub-class of quadratic functions.
>
> > __Q5:__ What is the difference between O(.) and \Omega(.)? Why the need for two different notations? Also, Thm 1 uses \Theta(.), what is that?
>
>
> __A5:__ Thank you for your question, we will add this in the notations section.
>
> - $f(n) = \mathcal{O}(g(n))$ if there exist $C >0$ and $n_0$ such that for all $n \geq n_0$ the following inequality holds, $0\leq f(n) \leq C g(n)$;
> - $f(n) = \Omega(g(n))$ if there exist $c>0$ and $n_0$ such that for all $n \geq n_0$ the following inequality holds, $0\leq cg(n) \leq f(n) $;
> - $f(n) = \Theta(g(n))$ if there exist $ C> 0$ and $c>0$ and $n_0$ such that for all $n \geq n_0$ the following inequality holds, $0\leq cg(n ) \leq f(n) \leq C g(n) $;
>
> For more details, we refer to the book: Cormen, Thomas H., Charles E. Leiserson, Ronald L. Rivest, and Clifford Stein. Introduction to algorithms. MIT press, 2022.

---

> > ### Comment · Reviewer_JJgW · 2025-08-05
> >
> > I thank the authors for their detailed responses. I encourage them to include the numerical results discussed in their rebuttal in the final submission. I am maintaining my current score.

---

### Official Review · Reviewer_oE5T · 2025-07-03

**Clarity:** 3
**Significance:** 2
**Originality:** 2
**Rating:** 4
**Confidence:** 4

**Summary:**

In this submission, the authors present a theoretical analysis of second-order optimization in the presence of heavy-tailed noise, addressing the common issue that such methods often underperform and lack guarantees in this setting. They consider scenarios where stochastic gradients and Hessians have only bounded first- and second-order moments, and derive tight lower bounds on the sample complexity for any second-order method. To complement their analysis, the authors propose a normalized stochastic gradient descent algorithm that leverages second-order information and matches the established lower bounds. They also introduce a novel gradient and Hessian clipping technique, along with high-probability upper bounds to support its effectiveness.

**Questions:**

1. From assumptions one to four, this analysis seems to be independent of the convexity of the objective function. All results are applicable in the non-convex setting. If we add the assumption of convexity or even strong convexity, how will these convexity condition improve the lower bound and the sample complexity of the proposed algorithms? All assumptions from this submission are relevant to the smoothness or Lipshiz continuity of both gradients and Hessian. If we add some lower bounds for the smallest eigenvalue of the Hessian, how will the the number of oracle queries required to find an ε-stationary point be reduced? Will this be some future research direction?

2. I'm not sure whether the proposed Normalized stochastic gradient decent with Hessian correction (with gradient clipping and Hessian clipping) belong to the class of second-order optimization algorithms. In fact this proposed algorithm utilize the second-order Hessian matrix information, but in general second-order optimization algorithm correspond to Newton type methods such as Newton's method, quasi-Newton method (BFGS, DFP ,SR1) or L-BFGS with iterations like: $x_{t + 1} = x_t - B_t \nabla{f(x_t)}$ where $B_t$ is some matrix that approximates the exact Hessian matrix $\nabla^2{f(x_t)}$. This Hessian approximation matrix $B_t$ is usually constructed using some first-order gradient informations.  However, the proposed variant SGD method has no such iteration updating forms. Hence, strictly speaking, the proposed Normalized SGD with Hessian correction belong to the first-order optimization algorithms which utilized some second-order Hessian queries. This method don't belong to the formal "second-order" optimization class.

3. From Theorem 2 and Theorem 3, both the step size $\gamma$ from the normalized SGD and the clipping threshold $\lambda$ from the clipping technique requires the estimate the initial sub-optimality $\delta = f(x_0) - f(x_*)$. How to estimate this initial sub-optimality in practice? Theorem 2 and 3 presented that $\gamma$ and $\lambda$ need to be set by values of this initial sub-optimality $\delta$ to reach the corresponding sample complexity. Hence, the proposed methods are impractical since it is in general very difficult to estimate the initial sub-optimality accurately.

**Ethical Concerns:**

["NO or VERY MINOR ethics concerns only"]

**Final Justification:**

The rebuttal results of numerical experiments solve my concerns. I raised the score from 3 to 4.

**Limitations:**

The authors have presented all the limitations of this work in the last section of the paper.

**Paper Formatting Concerns:**

No Paper Formatting Concerns

**Quality:**

2

**Strengths And Weaknesses:**

This paper makes several high-quality theoretical contributions, summarized as follows:

1. It establishes a lower bound on the sample complexity for any second-order stochastic optimization algorithm under the p-BCM noise model, showing that using higher-order derivatives does not necessarily improve the convergence rate under the same noise assumptions.

2. It proposes a stochastic gradient descent method that leverages second-order Hessian information and proves that its sample complexity matches the previously established lower bound.

3. It introduces a gradient and Hessian clipping technique and demonstrates that this method can achieve near-optimal sample complexity with high confidence.

These theoretical advances contribute significantly to the optimization community’s understanding of second-order stochastic methods in high-noise settings.

The main weakness of this submission is the lack of sufficient empirical results. The paper should include more numerical experiments evaluating the proposed Normalized SGD with Hessian correction algorithm and the gradient and Hessian clipping methods. Presenting empirical performance would help verify whether these novel algorithms align with the theoretical predictions.

For instance, since the normalized SGD with Hessian correction combined with clipping requires a clipping threshold parameter $\lambda$, the authors should explore the impact of varying $\lambda$ values on the algorithm’s performance. This would help assess whether the clipping technique enhances robustness. Incorporating such numerical experiments would significantly strengthen the paper and, if the empirical results support the theory, greatly increase its publication potential.

---

> ### Author Rebuttal · Authors · 2025-07-30
>
> Dear Reviewer oE5T,
>
> Thank you for your detailed and thoughtful review. We sincerely appreciate your recognition of our theoretical contributions, including the lower bound under the p-BCM noise model, the matching upper bound via our second-order algorithm, and the development of the gradient and Hessian clipping technique that achieves near-optimal sample complexity with high probability.
>
> We would like to address your main concern regarding the empirical evaluation:
>
>
> >__Q1:__ The main weakness of this submission is the lack of sufficient empirical results. The paper should include more numerical experiments evaluating the proposed Normalized SGD with Hessian correction algorithm and the gradient and Hessian clipping methods. Presenting empirical performance would help verify whether these novel algorithms align with the theoretical predictions.
>
>
> __A1:__ During the rebuttal period, we conducted several numerical experiments that are not included in the current version of the paper but will be added in the final revision. Due to NeurIPS policy, we are unable to include figures; however, we describe the results below.
> We solved a simple quadratic problem, $\frac{1}{2}\|x\|^2$, in dimension $d = 10$, with synthetic noise sampled from a symmetrized Pareto distribution. We considered both a fixed heavy-tailed regime with tail index $p = 1.1$ and a variable tail index ranging from $1.1$ to $2.0$. We start each algorithm from the same point sampled uniformly at random from the standard normal distribution.
> Our empirical analysis consists of three experimental settings:
>
> __1. Effect of Clipping Level on Iteration Complexity:__
> We ran Algorithm 2 (Clip NSGDHess) with $\lambda_h = \lambda$ until it reached the target accuracy of $\|\nabla F(x)\| \leq 3/2$. The stepsize and momentum parameters were set to $\gamma = 0.01$ and $\alpha = 0.2$. Clipping levels were varied from $10^{-16}$ to $10^3$ using a multiplicative step of 10. The results are shown below:
> | $\lambda$        | $10^{-16}$ | $10^{-8}$ | $10$ | $10^{2}$ | $10^{3}$ |
> |------------------|------------|-----------|------|----------|----------|
> | Iterations       | $632$        | $471$       | $\mathbf{321}$ | $562$      | $1506$     |
>
> Table 1: Intermediate clipping ($\lambda = 10$) yields the lowest iteration complexity. Extremely small or large values lead to slower convergence. The proposed algorithm can gracefully tolerate very small values of clipping parameter $\lambda$, but using values larger than $\lambda = 10^3$ leads to very slow convergence. This observation motivates the need for gradient and hessian clipping.
>
> __2.Comparison with Clip NSGDM under Varying Tail Index:__
> We compared Clip NSGDHess with Clip NSGDM (as proposed in [2]). For Clip NSGDM, we used the stepsize $\gamma = T^{-\frac{2p-1}{3p-2}}$ and momentum $\alpha = T^{-\frac{p}{3p-2}}$ according to theory. For Clip NSGDHess, both parameters were set to $T^{-\frac{p}{2p-1}}$. We fixed $\lambda = 0.5$, $\bar{\lambda}_h = 0.05$, and $T = 4000$ iterations.
>
> | Tail index, $p$ | Iteration Complexity (Clip NSGDM) | Iteration Complexity (Clip NSGDHess) |
> |----------------|--------------------------------------------|--------------------------------------|
> | $1.2$            | $2476$                                       | $2128$                                 |
> | $1.6$            | $1183$                                       | $726 $                                 |
> | $2.0$            | $877$                                        | $444$                                  |
>
>
> Table 2: Clip NSGDHess consistently outperforms Clip NSGDM, with the gap increasing for smaller $p$, i.e., heavier-tailed noise. Visually the picture looks similar to theoretical complexities in Figure 1 of our original submission.
>
> [2] Cutkosky, Ashok, and Harsh Mehta. "High-probability bounds for non-convex stochastic optimization with heavy tails." Advances in Neural Information Processing Systems 34 (2021): 4883-4895.
>
>
>
>
> > __Q2:__ From assumptions one to four, this analysis seems to be independent of the convexity of the objective function. All results are applicable in the non-convex setting. If we add the assumption of convexity or even strong convexity, how will these convexity condition improve the lower bound and the sample complexity of the proposed algorithms?
>
> __A2:__ In the convex case, under bounded variance ($p=2$) the upper and lower bounds were derived, e.g., in [1]. It turns out that when the function is convex, the use of second-order information does not improve the stochastic gradient complexity $O(\varepsilon^{-2})$. Therefore, we do not expect further improvement for other $p\in (1, 2)$ for our method as well. However, the situation in our non-convex setting is different and the sample complexity can be reduced uniformly for all $p\in (1, 2]$ using the second-order information.
>
> [1] Artem Agafonov, Dmitry Kamzolov, Alexander Gasnikov, Ali Kavis, Kimon Antonakopoulos, Volkan Cevher, Martin Takáč. Advancing the lower bounds: An accelerated, stochastic, second-order method with optimal adaptation to inexactness. ICLR 2024.
>
> > __Q3:__ All assumptions from this submission are relevant to the smoothness or Lipshiz continuity of both gradients and Hessian. If we add some lower bounds for the smallest eigenvalue of the Hessian, how will the the number of oracle queries required to find an $\varepsilon$-stationary point be reduced? Will this be some future research direction?
>
> __A3:__ Since we assume $L$-Lipschitz continuous gradient (smoothness) and the function is twice-differentiable, we already have a lower bound for the smallest eigenvalue of the Hessian equal to $\ell = -L$. Therefore, such assumption does not help improve the complexity in terms of the $\varepsilon$. However, one can consider a more refined setting where the smallest eigenvalue is $\ell > -L$ and improve the sample complexity in terms of dependence on $L$. We leave such refinement for future work.
>
>
> > __Q4:__  I'm not sure whether the proposed Normalized stochastic gradient decent with Hessian correction (with gradient clipping and Hessian clipping) belong to the class of second-order optimization algorithms. In fact this proposed algorithm utilize the second-order Hessian matrix information, but in general second-order optimization algorithm correspond to Newton type methods such as Newton's method, quasi-Newton method (BFGS, DFP ,SR1) or L-BFGS with iterations like: $x_{t+1}  = x_t - B_t \nabla f(x_t)$ where $B_t$ is some matrix that approximates the exact Hessian matrix $\nabla^2 f(x_t)$. This Hessian approximation matrix  is usually constructed using some first-order gradient informations. However, the proposed variant SGD method has no such iteration updating forms. Hence, strictly speaking, the proposed Normalized SGD with Hessian correction belong to the first-order optimization algorithms which utilized some second-order Hessian queries. This method don't belong to the formal "second-order" optimization class.
>
>
> __A4:__ Thank you for your thoughtful comment. We respectfully clarify that the use of second-order information is the defining characteristic of second-order optimization methods, and such methods are not necessarily restricted to the Newton-type update form $x_{t+1}  = x_t - B_t \nabla f(x_t)$. For instance, cubic Newton methods and trust-region methods are widely recognized as second-order algorithms, yet they do not conform to this canonical iteration. Similarly, our proposed Normalized SGD with Hessian correction explicitly utilizes Hessian information to adapt the updates, thereby leveraging second-order curvature information.
> Therefore, while our method differs structurally from Newton and quasi-Newton schemes, it naturally belongs to the broader class of second-order optimization methods.
>
>
> > __Q5:__  From Theorem 2 and Theorem 3, both the step size $\gamma$ from the normalized SGD and the clipping threshold $\lambda$ from the clipping technique requires the estimate the initial sub-optimality $\delta = f(x) - f(x_*)$. How to estimate this initial sub-optimality in practice? Theorem 2 and 3 presented that $\gamma$ and $\lambda$ need to be set by values of this initial sub-optimality \delta to reach the corresponding sample complexity. Hence, the proposed methods are impractical since it is in general very difficult to estimate the initial sub-optimality accurately.
>
> __A5:__ We can propose two strategies. If a lower bound on $F(x_*) \geq F^{\text{LB}}$ is available (e.g., it is zero) we can replace $\delta$ with $F(x) - F^{\text{LB}}$, where $F(x)$ can be estimated, e.g., using a mini-batch. In general any upper bound on $\delta$ suffices. Another valid strategy that will work for Algorithm 1 is to set step-size $\gamma = \sqrt{\frac{ \alpha^{\frac{1}{p}}}{ T}}$, momentum $\alpha = \left(\frac{1}{T}\right)^{\frac{p}{2p-1}}$ if $p$ is known or $\gamma = \sqrt{\frac{ \alpha^{\frac{1}{2}}}{ T}}$, momentum $\alpha = \left(\frac{1}{T}\right)^{\frac{2}{3}}$ if $p$ is unknown. Following the proof strategy similar to (Hübler et al, 2024), we can establish convergence of our algorithm. If $p$ is known the convergence rate of the algorithm will have the same dependence on $T$ and $p$ as Theorem 2, i.e., $\mathcal{O}(T^{-\frac{p-1}{2p-1}})$ and if $p$ is unknown, the method will still converge but slower, i.e., $\mathcal{O}(T^{-\frac{2(p-1)}{3p}})$.
>
> Florian Hübler, Ilyas Fatkhullin, Niao He. From Gradient Clipping to Normalization for Heavy Tailed SGD. AISTATS 2025.

---

### Official Review · Reviewer_GxUC · 2025-07-03

**Clarity:** 4
**Significance:** 4
**Originality:** 4
**Rating:** 5
**Confidence:** 4

**Summary:**

This paper studies nonconvex stochastic optimization for smooth functions where both stochastic gradients and Hessians have $p$-th bounded central moment for $p \in (1, 2]$. The authors establish minmax lower bounds on the sample complexity of second order stochastic methods for the considered problem class and zero-respecting algorithms. A near optimal algorithm, i.e., Normalized SGD with Hessian correction, is proposed and shown to achieve in-expectation upper bounds that almost matches with the lower bound. In addition, the, a clipped variant of the previous algorithm, i.e., Normalized SGD with Hessian correction and clipping, is shown to achieve nearly optimal high-probability convergence.

**Questions:**

1. In line 160, is it $q$-th order?
2. In definition 1, can the authors provide some examples on algorithms that belong to the defined algorithm family?

**Ethical Concerns:**

["NO or VERY MINOR ethics concerns only"]

**Final Justification:**

I have read the reviews from other reviewers and author's rebuttal. I decided to keep my original score.

**Limitations:**

N.A.

**Paper Formatting Concerns:**

N.A.

**Quality:**

4

**Strengths And Weaknesses:**

Strengths:
1. This paper for the first time proposes SOSO methods to address heavy-tailed noise.
2. The minmax lower bound is shown to be tight in several special cases.
3. Near-optimal algorithms are designed to match the lower bounds in the in-expectation, and high-probability sense with logarithmic failure probability dependence, respectively.

Weakness:
1. There is a lack of compelling numerical or analytical examples showcasing the advantages of SOSO over first-order methods.

---

> ### Author Rebuttal · Authors · 2025-07-30
>
> Dear Reviewer GxUC,
>
> Thank you very much for your detailed and encouraging review. We appreciate your positive assessment of the paper’s originality, theoretical rigor, and clarity. We are glad that you found the contributions—including the tight minimax lower bound and the near-optimal SOSO algorithms under heavy-tailed noise—novel and significant. Below, we address your thoughtful comments regarding numerical validation, clarify technical points (such as the algorithm family in Definition 1), and correct minor inconsistencies in the manuscript.
>
> > __Q1:__ In line 160, is it $q$-th order?
>
>
> __A1:__ Yes, this is a typo, thanks for catching this.
>
>
> >__Q2:__  In definition 1, can the authors provide some examples on algorithms that belong to the defined algorithm family?
>
>
> __A2:__ Examples of algorithms satisfying definition 1 include many common algorithm. For example, SGD (with and without momentum), stochastic BFGS and L-BFGS, Stochastic Newton’s method (with and without cubic regularization), first and second-order trust-region method. To support our claim we refer to Carmon et al., 2019 (page 80 in the published version), see also Arjevani et al., 2020. We will add a remark about this important question in our revision.
>
> Y. Carmon, J. C. Duchi, O. Hinder, and A. Sidford. Lower bounds for finding stationary points I. Mathematical Programming, May 2019.
>
> Yossi Arjevani, Yair Carmon, John C. Duchi, Dylan J. Foster, Ayush Sekhari, Karthik Sridharan. Second-Order Information in Non-Convex Stochastic Optimization: Power and Limitations. COLT 2020.

---

> > ### Comment · Reviewer_GxUC · 2025-08-07
> >
> > I thank the author for answering my questions. It is appreciated.

---

### Decision · Program_Chairs · 2025-09-17

**Decision:**

Accept (poster)

**Comment:**

The paper is an analysis of second-order stochastic optimization under the assumptions of heavy tailed stochastic gradients and hessians: it proposes a normalized SGD algorithms that achieves the established lower bounds on the sample complexity, and a novel gradient and Hessian clipping strategy to cope with large deviation instabilities, also achieving nearly-optimal performances. The main novelty of the paper consists in a sound and rigorous theoretical analysis of this relevant set-up, which has been appreciated for its clarity and originality. I recommend however the authors to integrate the text with the numerical experiments discussed with the Reviewers, as one of the weaknesses pointed out has been in fact the minimal experimental validation. Nevertheless, the relevance of the topic — essential for the construction of more realistic theoretical setup and more effective algorithmic solutions — makes the paper a valid and impactful contribution, and therefore I lean towards its acceptance.